# Content Provider Dynamics and Coordination in Recommendation Ecosystems

**Omer Ben-Porat**
Technion
Haifa 32000 Israel
omerbp@campus.technion.ac.il

**Itay Rosenberg**
Technion
Haifa 32000 Israel
itayrose@campus.technion.ac.il

**Moshe Tennenholtz**
Technion
Haifa 32000 Israel
moshet@ie.technion.ac.il

## Abstract

Recommendation Systems like YouTube are vibrant ecosystems with two types of users: Content consumers (those who watch videos) and content providers (those who create videos). While the computational task of recommending relevant content is largely solved, designing a system that guarantees high social welfare for *all* stakeholders is still in its infancy. In this work, we investigate the dynamics of content creation using a game-theoretic lens. Employing a stylized model that was recently suggested by other works, we show that the dynamics will always converge to a pure Nash Equilibrium (PNE), but the convergence rate can be exponential. We complement the analysis by proposing an efficient PNE computation algorithm via a combinatorial optimization problem that is of independent interest.

## 1 Introduction

Recommendation systems (RSs hereinafter) play a major role in our life nowadays. Many modern RSs, like YouTube, Medium, or Spotify, recommend content created by others and go far beyond recommendations. They are vibrant ecosystems with *multiple stakeholders* and are responsible for the well-being of all of them. For example, in the online publishing platform Medium, the platform should be profitable; suggest relevant content to the content consumers (readers); and support the content providers (authors). In light of this ecosystem approach, research on RSs has shifted from determining consumers' taste (e.g., the Netflix Prize challenge [9, 25]) to other aspects like fairness, ethics, and long-term welfare [5, 29, 31, 35, 37, 40–42, 44].

Understanding content providers and their utility[1] is still in its infancy. Content providers produce a constant supply of content (e.g., articles in Medium, videos on YouTube), and are hence indispensable. Successful content providers rely on the RS for some part of their income: Advertising, affiliated marketing, sponsorship, and merchandise; thus, unsatisfied content providers might decide to provide a different type of content or even abandon the RS. To illustrate, a content provider who is unsatisfied with her exposure, which is heavily correlated with her income from the RS, can switch to another type of content or seek another niche. Such downstream effects are detrimental to content consumer satisfaction because they *change the available content* the RS can recommend. The synergy between content providers and consumers is thus fragile, and solidifying one side solidifies the other.

In this paper, we investigate the *dynamics* of RSs using a stylized model in which content providers are strategic. Content providers obtain utility from displays of their content and are willing to change the content they offer to increase their utility. These fluctuations change not only the utility of the providers but also the social welfare of the consumers, defined as the quality of their proposed content. We show that the provider dynamics always converges to a stable point (namely, a pure Nash equilibrium), but the convergence time may be long. This observation suggests a more centralized approach, in which the RS coordinates the providers, and leads to fast convergence.

While our model is stylized, we believe it offers insights into more general, real-world RSs. The game-theoretic modeling allows counterfactual reasoning about the content that could-have-been-generated, which is impossible to achieve using existing data-sets and small online experiments. Our analysis advocates increased awareness to content providers and their incentives, a behavior that rarely exists these days in RSs.[2]

**Our contribution**    We explore the ecosystem using the following game-theoretic model, and use the blogging terminology to simplify the discussion. We consider a set of players (i.e., content providers), each selects a topic to write from a predefined set of topics (e.g., economics, sports, medieval movies, etc.). Each player has a *quality* w.r.t. each topic, quantifying relevance and attractiveness of that author's content if she writes on that topic, and a *conversion rate*. Given a selection of topics (namely, a strategy profile), the RS serves users who consume content. All queries concerned with a topic are modeled as the demand for that topic. The utility every player obtains is the sum of displays her content receives (affected by the demand for topics and the operating RS) multiplied by the conversion rate. The game-theoretic model we adopt in service is suggested by Ben Basat et al. [4] and is well-justified by later research [8, 42].

Technically, we deal with the question of reaching a stable point—a point in which none of the players can deviate from her selected topic and increase her utility. We are interested in the convergence time and the welfare of the system in these stable points. We first explore the decentralized approach: Better-response learning dynamics (see, e.g., [16, 21]), in which players asynchronously deviate to improve their utility (an arbitrary player to an arbitrary strategy, as long as she improves upon her current utility). We show that every better-response dynamic converges, thereby extending prior work [8]. Through a careful recursive construction, we show a negative result: The convergence time can be exponential in the number of topics. Long convergence time suggests a different approach. We consider the scenario in which the RS could act centrally, and support the process of *matching* players with topics. We devise an algorithm that computes an equilibrium fast (roughly squared in the input size). To solve this computational challenge, which is a mixture of matching and load-balancing, we propose a novel combinatorial optimization problem that is of independent interest.

Conceptually, we offer a qualitative grounding for the advantages of coordination and intervention[3] in the content provider dynamics. Our analysis relies on the assumption of complete knowledge of all model parameters, in particular the qualities. While unrealistic in practice, we expect that incomplete information will only exacerbate the problems we address. The main takeaway from this paper is that RSs are not self-regulated markets, and as much as suggesting authors topics to write on can lead to a significant increase in the system's stability. We discuss some practical ways of reaching this goal in Section 5.

**Related work**    Strikingly, content provider welfare and their fair treatment were only suggested very recently in the Recommendation Systems and Information Retrieval communities [12, 14, 18, 35, 40, 46]. All of these works do not model the incentives of content providers explicitly, and consequently cannot offer a what-if analysis like ours.

Our model is similar to those employed in several recent papers [4, 5, 7, 8, 30]. Ben-Porat et al. [8] study a model that is a special case of ours, and show that every learning dynamic converges. Our Theorem 1 recovers and extends their convergence results. Moreover, unlike this work, they do not address convergence time, social welfare, and centralized equilibrium computation. Other works [5, 7, 30] aim to design recommendation mechanisms that mitigate strategic behavior and

lead to long-term welfare. On the negative side, their mechanisms might knowingly recommend inferior content to some consumers. We see their work as parallel to ours, as in this work we focus on the prevailing recommendation approach—recommending the best-fitting content. We suggest that a centralized approach, in which the RS orchestrates the player-topic matching, can significantly improve the time until the system reaches stability (in the form of equilibrium). Furthermore, we envision that our approach can also lead to high social welfare, as we discuss in Section 5.

More broadly, an ever-growing body of research deals with fairness considerations in Machine Learning [15, 17, 36, 38, 45]. In the context of RSs, a related line of research suggests fairer ranking methods to improve the overall performance [11, 26, 43]. For example, Yao and Huang [43] propose metrics mitigating discrimination in collaborative-filtering methods that arise from learning from historical data. Despite not always being explicit, the ultimate goal of fairness imposition is to achieve long-term welfare [28]. Our paper and analysis share a similar flavor: To achieve high stability via faster convergence, RSs should coordinate the process of content selection.

## 2 Model

We consider the following recommendation ecosystem, where for concreteness we continue with the blog authors[4] example. There is a set of authors $\mathcal{P}$, each owning a blog. We further assume that each blog is concerned with a single topic, from a predefined topic set $\mathcal{T}$. We assume $\mathcal{P}$ and $\mathcal{T}$ are finite, and denote $|\mathcal{P}| = P$ and $|\mathcal{T}| = T$. The strategy space of each player is thus $\mathcal{T}$; she selects the topic she writes on. A pure strategy profile is a tuple $\boldsymbol{a} = (a_1, \ldots a_P)$ of topic selections, where $a_j$ is the topic selected by author $j$.

For every author $j$ and topic $k$, there is a *quality* that quantifies the relevance and attractiveness of $j$'s blog if she picks the topic $k$. We denote by $\mathcal{Q}$ the quality matrix, for $\mathcal{Q} \in [0,1]^{P \times T}$. The RS serves users who consume content. We do not distinguish individual consumers, but rather model the need for content as a demand for each topic. A demand distribution $\mathcal{D}$ over the topics $\mathcal{T}$ is publicly known, where we use $\mathcal{D}(k)$ to denote the demand mass for topic $k \in \mathcal{T}$. W.l.o.g., we assume that $\mathcal{D}(1) \geq \mathcal{D}(2) \geq \ldots \geq \mathcal{D}(m)$.

The recommendation function $\mathcal{R}$ matches demand with available blogs. Given the demand for topic $k$, a strategy profile $\boldsymbol{a}$, and the quality $\mathcal{Q}$ of the blogs for the selected topics in $\boldsymbol{a}$, the recommendation function $\mathcal{R}$ recommends content, possibly in a randomized manner. It is well-known that content consumers pay most of their attention to highly ranked content [13, 22, 24, 27]; therefore, we assume for simplicity that $\mathcal{R}$ recommends one content solely. For ease of notation, we denote $\mathcal{R}_j(\mathcal{Q}, k, \boldsymbol{a})$ as the probability that author $j$ is ranked first under the distribution $\mathcal{R}(\mathcal{Q}, k, \boldsymbol{a})$ (or rather, author $j$'s content is ranked first). While blog readers admire high-quality recommended blogs, blog authors care for payoffs. As described in Section 1, authors draw monetary rewards from attracting readers in various ways. We model this payoff abstractly using a *conversion matrix* $\mathcal{C}, \mathcal{C} \in [0,1]^{P \times T}$. We assume that every blog reader grants $\mathcal{C}_{j,k}$ monetary units to author $j$ when she writes on topic $k$. For example, if author $j$ only cares for exposure, namely the number of impressions her blog receives, then $\mathcal{C}_{j,k} = 1$ for every $k \in \mathcal{T}$. Alternatively, if author $j$ cares for the engagement of readers in her blog, then the conversion $\mathcal{C}_{j,k}$ should be somewhat correlated with the quality $\mathcal{Q}_{j,k}$. We will return to these two special cases later on, in Subsection 3.1. The *utility* of author $j$ under a strategy profile $\boldsymbol{a}$ is given by

$$\mathcal{U}_j(\boldsymbol{a}) \stackrel{\text{def}}{=} \sum_{k \in \mathcal{T}} \mathbb{1}_{a_j=k} \cdot \mathcal{D}(k) \cdot \mathcal{R}_j(\mathcal{Q}, k, \boldsymbol{a}) \cdot \mathcal{C}_{j,k}. \tag{1}$$

Overall, we represent a game as a tuple $\langle \mathcal{P}, \mathcal{T}, \mathcal{D}, \mathcal{Q}, \mathcal{C}, \mathcal{R}, \mathcal{U} \rangle$, where $\mathcal{P}$ is the authors, $\mathcal{T}$ is the topics, $\mathcal{D}$ is the demand for topics, $\mathcal{Q}$ and $\mathcal{C}$ are the quality and conversion matrices, $\mathcal{R}$ is the recommendation function, and $\mathcal{U}$ is the utility function.

**Recommending the Highest Quality Content**   In this paper, we focus on the RS that recommends blogs of the highest quality, breaking ties randomly. Such a behavior is intuitive and well-justified in the literature [3, 10, 23, 39]. More formally, let $B_k(\boldsymbol{a})$ denote the highest quality of a blog written on topic $k$ under the profile $\boldsymbol{a}$, i.e., $B_k(\boldsymbol{a}) \stackrel{\text{def}}{=} \max_{j \in \mathcal{P}} \{\mathbb{1}_{a_j=k} \cdot \mathcal{Q}_{j,k}\}$. Furthermore, let $H_k(\boldsymbol{a})$ denote the set of authors whose documents have the highest quality among those who write on topic $k$ under

$\boldsymbol{a}$, $H_k(\boldsymbol{a}) \stackrel{\text{def}}{=} \{j \in \mathcal{P} \mid \mathbb{1}_{a_j=k} \cdot \mathcal{Q}_{j,k} = B_k(\boldsymbol{a})\}$. The recommendation function $\mathcal{R}^{\text{top}}$ is therefore defined as

$$\mathcal{R}_j^{\text{top}}(\mathcal{Q}, k, \boldsymbol{a}) \stackrel{\text{def}}{=} \begin{cases} \frac{1}{|H_k(\boldsymbol{a})|} & j \in H_k(\boldsymbol{a}) \\ 0 & \text{otherwise} \end{cases}.$$

Consequently, we can reformulate the utility function from Equation (1) in the following succinct form,[5]

$$\mathcal{U}_j(\boldsymbol{a}) \stackrel{\text{def}}{=} \sum_{k \in \mathcal{T}} \mathbb{1}_{a_j=k} \cdot \frac{\mathcal{D}(k)}{|H_k(\boldsymbol{a})|} \cdot \mathcal{C}_{j,k}. \tag{2}$$

From here on, since $\mathcal{R}^{\text{top}}$ and $\mathcal{U}$ are fully determined by the rest of the objects, we omit them from the game representation; hence, we represent every game by the more concise tuple $\langle \mathcal{P}, \mathcal{T}, \mathcal{D}, \mathcal{Q}, \mathcal{C} \rangle$.

**Quality-Conversion Assumption** Throughout the paper, we make the following Assumption 1 about the relation between quality and conversion.

**Assumption 1.** *For every topic $k \in \mathcal{T}$ and every two authors $j_1, j_2 \in \mathcal{P}$,*

$$\mathcal{Q}_{j_1,k} \geq \mathcal{Q}_{j_2,k} \Rightarrow \mathcal{C}_{j_1,k} \geq \mathcal{C}_{j_2,k}.$$

Intuitively, Assumption 1 implies that quality and conversion are correlated given the topic. For every topic $k$, if authors $j_1$ and $j_2$ write on topic $k$ and $j_1$'s content has a weakly better quality, then $j_1$'s content has also a weakly better conversion. This assumption plays a crucial role in our analysis; we discuss relaxing it in Section 5.

**Solution Concepts** The social welfare of the readers is the average weighted quality. Formally, given a strategy profile $\boldsymbol{a}$,

$$SW(\boldsymbol{a}) \stackrel{\text{def}}{=} \sum_{k \in \mathcal{T}} \mathcal{D}(k) \sum_{j \in \mathcal{P}} \mathcal{R}_j(\mathcal{Q}, k, \boldsymbol{a}) \mathcal{Q}_{j,k}. \tag{3}$$

As the recommendation function $\mathcal{R}^{\text{top}}$ always recommends the highest quality content, we can have the following more succinct representation of social welfare, $SW(\boldsymbol{a}) = \sum_{k \in \mathcal{T}} \mathcal{D}(k) B_k(\boldsymbol{a})$. However, social welfare maximization does not concern author utility. Authors may be willing to deviate from the socially optimal profile if such a deviation is beneficial in terms of utility. Consequently, we seek stable solutions, as captured by the property of pure Nash equilibrium (hereinafter PNE). We say that a strategy profile $\boldsymbol{a}$ is a PNE if for every author $j$ and topic $k$, $\mathcal{U}_j(\boldsymbol{a}) \geq \mathcal{U}_j(\boldsymbol{a}_{-j}, k)$, where $\boldsymbol{a}_{-j}$ is the tuple obtained by deleting the $j$'s entry of $\boldsymbol{a}$. It is worth noting that while mixed Nash equilibrium is guaranteed to exist in finite games, a PNE generally does not exist in games. However, as we show later on, it always exists in our class of games.

**Example** To clarify our notation and setting, we provide the following example. Consider a game with two players ($P = 2$), two topics ($T = 2$) and the demand distribution $\mathcal{D}$ such that $\mathcal{D}(1) = 3/5, \mathcal{D}(2) = 2/5$. Let the quality and conversion matrices be

$$\mathcal{Q} = \begin{pmatrix} 1 & 1/3 \\ 2/3 & 1/3 \end{pmatrix}, \qquad \mathcal{C} = \begin{pmatrix} 1/3 & 1 \\ 1/5 & 1 \end{pmatrix}.$$

Consider the strategy profile $(a_1, a_2) = (1, 1)$. Author 1 is more competent that author 2 on topic 1, since $\mathcal{Q}_{1,1} = 1 > \mathcal{Q}_{2,1} = \frac{2}{3}$; thus, the utility of author 1 under the profile $(1, 1)$ is $\mathcal{U}_1(1, 1) = \mathcal{D}(1) \cdot \mathcal{R}_1^{\text{top}}(\mathcal{Q}, 1, (1, 1)) \cdot \mathcal{C}_{1,1} = \frac{3}{5} \cdot 1 \cdot \frac{1}{3} = \frac{1}{5}$. On the other hand, author 2 gets $\mathcal{U}_2(1, 1) = \frac{3}{5} \cdot 0 \cdot \frac{1}{5} = 0$. Author 2 has a beneficial deviation: Under the profile $(1, 2)$, her utility is $\mathcal{U}_2(1, 2) = \frac{2}{5} \cdot 1 \cdot 1 = \frac{2}{5}$, while the utility of author 1 remains the same, $\mathcal{U}_1(1, 2) = \frac{1}{5}$. For the strategy profile $(2, 2)$, both authors have the same quality; thus, $\mathcal{R}_1^{\text{top}}(\mathcal{Q}, 2, (2, 2)) = \mathcal{R}_2^{\text{top}}(\mathcal{Q}, 2, (2, 2)) = \frac{1}{2}$. As for the utilities, $\mathcal{U}_1(2, 2) = \mathcal{U}_2(2, 2) = \frac{2}{5} \cdot \frac{1}{2} \cdot 1 = \frac{1}{5}$. Overall, we see that both $(1, 2)$ and $(2, 2)$ are PNEs, since the authors do not have beneficial deviations. However, the social welfare of these PNEs is different: $SW(1, 2) = \frac{3}{5} \cdot 1 + \frac{2}{5} \cdot \frac{1}{3} \approx 0.73$, yet $SW(2, 2) = \frac{3}{5} \cdot 0 + \frac{2}{5}\frac{1}{3} \approx 0.13$.

# 3 Decentralized Approach

In this section, we consider the prevailing, decentralized approach. Starting from an arbitrary profile, authors interact asynchronously, each improving her utility in every time step. Such dynamics is widely-known in the Game Theory literature as better-response dynamics (hereinafter, BRDs). Studying BRDs is a robust approach for assuring the environment reaches a stable point, while making minimal assumption on the information of the players. Two central questions about BRDs in games are a) whether *any* BRD converges; and b) what is the convergence rate. We show that the answer to the first question is in the affirmative. For the second question, we show through an intricate combinatorial construction a result of negative flavor: The convergence rate can be exponential in the number of topics $T$.

## 3.1 Better-Response Dynamic Convergence

Before we go on, we define BRDs formally. Given a strategy profile $\boldsymbol{a}$, we say that $a'_j \in \mathcal{T}$ is a *better response* of author $j$ w.r.t. $\boldsymbol{a}$ if $\mathcal{U}_j(\boldsymbol{a}_{-j}, a'_j) > \mathcal{U}_j(\boldsymbol{a})$. A BRD is a sequence of profiles $(\boldsymbol{a}^1, \boldsymbol{a}^2, \dots)$, where at every step $i + 1$ exactly one author better-responds to $\boldsymbol{a}^i$, i.e., there exists an author $j(i)$ such that $\boldsymbol{a}^{i+1} = (\boldsymbol{a}^i_{-j(i)}, a^{i+1}_{j(i)})$ and $\mathcal{U}_{j(i)}(\boldsymbol{a}^{i+1}) > \mathcal{U}_{j(i)}(\boldsymbol{a}^i)$. A BRD can start from any arbitrary profile, and include improvements of any arbitrary author at any arbitrary step (assuming she has a better response in that time step). If a BRD $\boldsymbol{a}^1, \dots, \boldsymbol{a}^l$ converges, namely no player can better respond to $\boldsymbol{a}^l$, then by definition $\boldsymbol{a}^l$ is a PNE.

Our goal is to show that every BRD of any game in our class of games converges. If there exists an infinite BRD, then it must contain cycles as the number of different strategy profiles is finite. Equivalently, nonexistence of improvement cycles suggests that any BRD will converge to a PNE [32]. General techniques for showing BRD convergence in games are rare, and are typically based on coming up with a potential function [6, 21, 34] or a natural lexicographic order [2, 19]. However, as already established by prior work [8, Proposition 1], our class of game does not fit into the category of an exact potential function; and a lexicographic order does not seem to arise naturally. Ben-Porat et al. [8] prove BRD convergence for two sub-classes of games: Games where $\mathcal{C}$ is identically 1, and games with $\mathcal{C} = \mathcal{Q}$. Interestingly, they prove BRD convergence for each sub-class separately using different arguments. We extend their technique to deal with any conversion matrix $\mathcal{C}$ that satisfies Assumption 1.

**Theorem 1.** *If a game $\mathcal{G}$ satisfies Assumption 1, then every BRD in $\mathcal{G}$ converges to a PNE.*

## 3.2 Rate of Convergence

We now move on to the second question proposed in the beginning of the section, which deals with convergence rate. The convergence rate is the worst-case length of any BRD. Recall that a BRD can start from a PNE and thus converge after one step, and hence the worst-case approach we offer here is justified.

Our next theorem lower bounds the worst case convergence rate by an exponential factor in the number of topics $T$. This result is illuminating as it shows that in the worst case, although convergence is guaranteed, it may not be reachable in feasible time.

**Theorem 2.** *Consider $P \geq 1$ and $T \geq 2$. There exist games satisfying Assumption 1 with $|\mathcal{P}| = P$ and $|\mathcal{T}| = T$, in which there are BRDs with at least $\left(\frac{T-2}{P} + 1\right)^P$ steps.*

**Proof sketch of Theorem 2.** The proof relies on a recursive construction. We construct a game and an improvement path with at least the length specified in the theorem. To balance rigor and intuition, we present here a special case of our general construction and defer the formal proof to the appendix.

Consider the game with $P = 3, T = 5, \mathcal{D}(k) = \frac{1}{5}$ for every $k \in \mathcal{T}$ and

$$\mathcal{Q} = \mathcal{C} = \begin{pmatrix} c & \boxed{2c \quad 3c \quad 4c \quad 5c} \\ c & \boxed{9c \quad 8c \quad 7c \quad 6c} \\ c & \boxed{10c \quad 11c \quad 12c \quad 13c} \end{pmatrix}$$

$$
\begin{array}{llllll}
a^1 = (2,1,1) & a^2 = (3,1,1) & a^3 = (4,1,1) & a^4 = (5,1,1) & a^5 = (5,5,1) & a^6 = (1,5,1) \\
a^7 = (2,5,1) & a^8 = (3,5,1) & a^9 = (4,5,1) & a^{10} = (4,4,1) & a^{11} = (1,4,1) & a^{12} = (2,4,1) \\
a^{13} = (3,4,1) & a^{14} = (3,3,1) & a^{15} = (1,3,1) & a^{16} = (2,3,1) & a^{17} = (2,2,1) & a^{18} = (1,2,1) \\
a^{19} = (1,2,2) & a^{20} = (1,1,2) & a^{21} = (3,1,2) & a^{22} = (4,1,2) & a^{23} = (5,1,2) & a^{24} = (5,5,2) \\
a^{25} = (1,5,2) & a^{26} = (3,5,2) & a^{27} = (4,5,2) & a^{28} = (4,4,2) & a^{29} = (1,4,2) & a^{30} = (3,4,2) \\
a^{31} = (3,3,2) & a^{32} = (1,3,2) & a^{33} = (1,3,3) & a^{34} = (1,1,3) & a^{35} = (4,1,3) & a^{36} = (5,1,3) \\
a^{37} = (5,5,3) & a^{38} = (1,5,3) & a^{39} = (4,5,3) & a^{40} = (4,4,3) & a^{41} = (1,4,3) & a^{42} = (1,4,4) \\
a^{43} = (1,1,4) & a^{44} = (5,1,4) & a^{45} = (5,5,4) & a^{46} = (1,5,4) & a^{47} = (1,5,5) & a^{48} = (1,1,5)
\end{array}
$$

Figure 1: A long improvement path for the instance in the proof sketch of Theorem 2.

for $c = \frac{1}{PT}$. The first column of the matrix, which is associated with the quality of topic 1, is identical for all authors. The snake-shape path in the matrix is always greater than the value $c$ in the first column, and is monotonically increasing (top-down). The immediate implications are a) odd players improve their quality when deviating to a topic with a greater index, while even players improve their quality when deviating to a topic with a smaller index (which is not topic 1); and b) every player is more competent than all the players that precede her on every topic but topic 1. The initial profile is $a^0 = (1, 1, \ldots, 1)$. We construct the BRD that appears in Figure 1.[6] It comprises three types of steps: Purple, green and yellow. In purple steps, author 1 deviates to a topic with a higher index. In yellow steps, author 2 deviates to the topic selected by author 1 (e.g., in $a^5$) or author 3 deviates to the topic selected by author 2 (e.g., in $a^{19}$). Green steps always follow yellow steps. In green steps, the author whose topic was selected in the previous step by an author with a higher index deviates back to topic 1 (e.g., author 1 in $a^6$ after author 2 selects topic 5 in $a^5$, or author 2 in $a^{20}$ after author 3 selects topic 2 in $a^{19}$).

In steps $a^1 - a^4$, only author 1 deviates (purple steps). This is also the recursive path in a game with author 1 solely (disregarding the entries of the other players). Then, in $a^5$, author 2 deviates to topic 5 (yellow). Since author 2 is more competent than author 1 in every topic (excluding topic 1), author 1's utility equals zero. Then, author 1 deviates to back topic 1 in $a^6$ (green). This goes on until step $a^{18}$—author 1 improves, author 2 ties, and author 1 returns to topic 1. Steps $a^1 - a^{18}$ comprise the recursive path for two players. Until step $a^{18}$, author 3 did not move. Then, in step $a^{19}$, author 3 deviates to topic 2. Author 3 is more competent than author 2, so in $a^{20}$ author 2 returns to topic 1. In steps $a^{21} - a^{32}$ authors 1 and 2 follow the same logic as before, but they overlook topic 2 (since author 3, who is more competent than both of them, selects it). In steps $a^{33} - a^{34}$ author 3 deviates to topic 3, and then author 2 returns to topic 1. In steps $a^{35} - a^{41}$ authors 1 and 2 follow the same logic as before, but they overlook both topics 2 and 3. The path continues similarly until we reach the profile $a^{48}$. Notice that the latter profile is not an equilibrium, but we end the path at this point for the sake of the analysis. This path is indeed exponential—for every step author $i$ makes, for $1 < i \leq 3$, author $i - 1$ makes at least twice as many (in fact, much more than that; see the formal proof for more details). □

Theorem 2 implies that there are BRDs of length $\left(\frac{T-2}{P} + 1\right)^P$, which is $O(\exp(T))$ for large enough $P$. Furthermore, if the number of topics $T$ and the authors $P$ are in the same order of magnitude, then length is also exponential in $P$.

## 4 Centralized Approach - Equilibrium Computation

To remedy the long convergence rate, in this section we propose an efficient algorithm for PNE computation. The algorithm is a matching application and relies on a novel graph-theoretic notion. To motivate the matching perspective, we reconsider social welfare (see Equation (3)) and neglect strategic aspects momentarily. We can find a social welfare-maximizing profile using the following matching reduction. We construct a bipartite graph, one side being the authors and the other side being the topics. The weight on each edge $(j, k)$ is $\mathcal{Q}_{j,k}\mathcal{D}(k)$, the quality author $j$ has on topic $k$ times the user mass on that topic. Notice that every author can only select one strategy (topic). Furthermore, for the purpose of social welfare maximization, it suffices to consider candidate profiles in which every topic is selected by at most one author. Consequently, a maximum weighted matching

of this graph corresponds to the social welfare maximizer. By using, e.g., the Hungarian algorithm, the problem of finding a social welfare-maximizing profile can be solved in $O(\max\{P, T\}^3)$.

However, equilibrium profiles and social welfare-maximizing profiles typically do not coincide (see the celebrated work on the Price of Anarchy [33]). The maximum matching that we proposed in the previous paragraph is susceptible to beneficial devotions; therefore, it is not stable in the equilibrium sense.[7] There exist many variants of stable matching in the literature, but virtually none fit the equilibrium stability we seek. In particular, the deferred acceptance algorithm [20] cannot be used since several players can select the same topic and thus the matching is not one-to-one. If we create several copies of the same topic (a common practice for the deferred acceptance algorithm), high-quality players would block low-quality authors matched to it (unlike several medical students with varying qualities that are matched to the same hospital). In the remainder of this section, we propose a sequential matching technique to compute a PNE. Our approach contributes to the matching literature and is based on the definition of *saturated sets*.

Due to our extensive use of graph theory in what follows, we introduce a few notational conventions. We denote a graph by $G = (V, E)$. For a subset $W \subset V$, the *induced sub-graph* $G[W]$ is the graph whose vertex set is $W$ and whose edge set consists of all the edges in $E$ that have both endpoints in $W$. We use the standard notation $N_G(W)$ to denote the neighbors of the vertices $W$ in the graph $G$. A matching $M$ in $G$ is a set of pairwise non-adjacent edges. For our application, we care mostly about bipartite graphs; thus, we denote $V = X \cup Y$. An $X$-saturating matching is a matching that covers every node in $X$. Hall's Marriage Theorem, a fundamental result in combinatorics, gives necessary and sufficient conditions for the existence of perfect matching. The theorem asserts that there exists an $X$-saturated matching in $G$ if and only if for every subset $W \subseteq X$, $|W| \leq |N_G(W)|$. In other words, the size of every subset in $X$ does not exceed the number of its neighbors. The essential property we use in the PNE algorithm is *saturated sets*.

**Definition 1** (Saturated set). *Let $G = (X \cup Y, E)$ be a finite bipartite graph. A set $W \subseteq X$ is called saturated if $|W| = |N_G(W)|$.*

Of course, this definition naturally extends beyond bipartite graphs. Furthermore, if for every other saturated set $W'$ it holds that $|W| \geq |W'|$, we say that $W$ is a maximum saturated set. Despite its striking simplicity, to the best of our knowledge, this notion of saturated sets did not receive enough attention in the CS literature (under this name or a different one), and is therefore interesting in its own right.

### 4.1  PNE Computation

We now turn to discuss the intuition behind Algorithm 1, which computes a PNE efficiently. By and large, Algorithm 1 can be seen as a best-response dynamic. It starts from a null profile (assigning all players to a factitious topic with zero user mass) and then determines the order of best-responding.

The input is the entire game description,[8] as described in Section 2. In Lines 1-5 we initialize the variables we use. $\tilde{\mathcal{T}}$ is the set of unmatched topics; $L_k$ is a lower bound on the *load* on topic $k$, namely the ongoing number of players we matched to it; $X, Y$ and $E$ are the elements of the bipartite graph $G$ ($Y$ stores the set of unmatched players); and $\boldsymbol{a}^*$ is a non-valid, empty profile that we construct as the algorithm advances. The for loop in Line 6 goes as follows. We first find the set of highest-quality players for every topic $k$, denoted $A_k$ (Line 7). These players can block the others from playing $k$ because their quality is higher, and thus we prioritize them in our sequential process. Afterwards, we set $k^*$ to be the most profitable topic under the current partial matching (Line 8). That is, for every topic $k$, we consider the set of most profitable players w.r.t. $k$ and their potential utility if matched to $k$. The term $\mathcal{D}(k)\mathcal{C}_{j,k}/L_k+1$ upper bounds the utility of every player $j \in A_k$ (see Equation (2)), in case we match $L_k + 1$ or more players to topic $k$ (we might increase the load $L_k$ in later iterations). We subsequently update $L_{K^*}$ in Line 9.

We now move to the bipartite graph $G$. In Line 10, we create a new node $x$, which is the $L_{k^*}$-copy of topic $k^*$ (we store this information about $x$). We add $x$ to the left side of $G$, $X$ (Line 11), and connect

___________________

[7]There are exceptions, of course. In degenerate cases where $\mathcal{Q}$ has no ties, the game is essentially a stable marriage problem.

[8]For the sake of illustration, we assume $P \leq T$. If that is not the case, we can add enough topics with zero mass $\mathcal{D}$ to achieve it. Noticeably, a PNE in the new game can be converted to a PNE in the original game.

---

**Algorithm 1:** PNE computation

**Input:** A game description $\langle \mathcal{P}, \mathcal{T}, \mathcal{D}, \mathcal{Q}, \mathcal{C} \rangle$

**Output:** A PNE $\boldsymbol{a}$

1   $\tilde{\mathcal{T}} \leftarrow \mathcal{T}$ // available topics

2   $\forall k \in \mathcal{T} : L_k \leftarrow 0$ // loads on topic

3   $X \leftarrow \emptyset, Y \leftarrow \mathcal{P}, E \leftarrow \emptyset$

4   $G \leftarrow (X \cup Y, E)$

5   $\boldsymbol{a}^* \leftarrow (\emptyset)^m$ // empty profile

6   **for** $t = 1 \dots P$

7     $\forall k \in \tilde{\mathcal{T}} : A_k \leftarrow \arg\max_{j \in Y} \mathcal{Q}_{j,k}$

8     set $k^* \in \arg\max_{k \in \tilde{\mathcal{T}}} \left\{ \max_{j \in A_k} \frac{\mathcal{D}(k)\mathcal{C}_{j,k}}{L_k+1} \right\}$

9     $L_{k^*} \leftarrow L_{k^*} + 1$

// for loop continues...

10    create a new node $x$ associated with topic $k^*$

11    $X.add(x)$

12    $E.add\left(\{(x,j) : j \in A_{k^*}\}\right)$

13    Let $W \subseteq X$ be the maximum saturated set in $G$

14    **if** $W \neq \emptyset$ **then**

15      find a maximum matching $M$ in $G[W \cup Y]$

16      $\forall j \in N_G(W) : a_j^* \leftarrow \text{Topic}(M(j))$

17      $Y.remove(N_G(W))$

18      $X.remove(W)$

19      $\tilde{\mathcal{T}}.remove(\text{Topics}(W))$ // see Line 10

20   **return** $\boldsymbol{a}^*$

---

$x$ to the players of $A_{k^*}$ in $Y$ (Line 12). Line 13 is the crux of the algorithm: We find a subset $W$ of $X$ that is the maximum saturated set. We will justify our use of the article *the* in the previous sentence later on, as well as describe the implications of having a saturated set in this dynamically constructed graph. If $W$ is empty, we continue to the next iteration of the for loop. But if $W$ is non-empty, we enter the if block in Line 14. We find a maximum matching $M$ in the induced graph $G[W \cup Y]$. We will later prove that $G[W \cup Y]$ satisfies Hall's marriage condition, and thus $|M| = |W| = |N_G(W)|$. In Line 16 we use $M$ to set the strategies of the players in $N_G(W)$: Every player $j \in N_G(W)$ is matched to the topic associated with the node $M(j) \in W$. In Lines 17-19 we remove the newly matched players $N_G(W)$ from $Y$, the topic copies $W$ from $X$, and the topics associated with $W$ from the set of unmatched topics $\tilde{\mathcal{T}}$. We repeat this process until all players are matched.

Let us explain the implications of having a non-empty saturated set in $G$. Focus on the first time a non-empty saturated set $W$ was found in Line 13, and denote the iteration index by $t'$. The set $W$ is composed of nodes associated with several topics (association in the sense we explain about Line 10); each one may have several copies. Importantly, every time we add a node $x$ to $X$ with an associated topic $k$, we increased the load $L_k$; hence, in iteration $t'$, $L_k$ accurately reflects the number of copies of $k$ in $X$. Furthermore, $k$ was selected for the $L_k + 1$ time, suggesting that it is more profitable than other topics. With a few more arguments, we show that all $L_k$ copies of $k$ must be in $W$. Crucially, if we match the players in $N_G(W)$ they cannot have beneficial deviations. We formalize this intuition via Theorem 3.

**Theorem 3.** *If the input game $\mathcal{G}$ satisfies Assumption 1, then Algorithm 1 returns a PNE of $\mathcal{G}$.*

We now move on to discuss its run-time. The only two lines that require a non-trivial discussion are Lines 13 and 15. As we describe in Lemma 1 below, finding the maximum saturated set includes finding a maximum matching, and thus we need not recompute it in Line 15. We therefore focus on the complexity of finding the saturated set in $G$ solely. The following Lemma 1 shows that as long as a bipartite $G$ satisfies Hall's marriage condition, we can find the maximum saturated set $W$ efficiently. Because of the independent interest in this combinatorial problem, we state it in its full generality.

**Lemma 1.** *Let $G = (V, E)$ be a bipartite graph that satisfies Hall's marriage condition. There exists an algorithm that finds the maximum saturated set of $G$ in time $O(\sqrt{|V|}|E|)$.*

The proof of this basic lemma appears in the appendix. The sketch of the proof is as follows. Let $G = (X \cup Y, E)$ be a graph satisfying Hall's marriage condition. We first compute a maximum matching $M$ of $G$. Since Hall's marriage condition holds, we are guaranteed that $M$ is an $X$-saturating matching. We then devise a technique to find whether a node $x \in X$ participates in at least one saturated set. We show that nodes participating in saturated sets are reachable from the set of unmatched nodes in $Y$ via a variation of alternating paths, and thus can be identified quickly. By the end of this procedure, we have a set $X' \subseteq X$ such that every $x \in X'$ participates in at least one saturated set. The last part is showing that under the marriage condition, every union of saturated sets is a saturated set. As a result, we conclude that $X'$ is the maximum saturated set. Using Lemma 1, we can bound the run-time of Algorithm 1.

**Corollary 1.** *Algorithm 1 can be implemented in running time of $O(P^{2.5} \cdot T)$.*

## 5 Discussion

With great effort, companies like Amazon turned the "you bought that, would you also be interested in this" feature into a significant source of revenue. In this paper, we suggest that a "you wrote this, would you also be interested in writing on that?" feature could be revolutionary as well—contributing to better social welfare of content consumers, as well as the utility of content providers. Such a policy could be implemented in practice by a direct recommendation to providers, or by a more moderate action like nudging content providers to experiment with a different set of contents. To support our vision of content provider coordination in RSs even further, we show in the appendix that the ratio between the social welfare of the best equilibrium and the worst equilibrium is unbounded. Indeed, such a coordination between content providers may lead to a significant lift in social welfare. More broadly, we note that maximizing the overall welfare of RSs with multiple stakeholders is an important challenge that goes way beyond this paper (see, e.g., [12]).

From a technical perspective, this work suggests a variety of open questions. First, the challenge of computing the social welfare-maximizing equilibrium is still open. Second, as we show in the appendix that if Assumption 1 does not hold, BRDs may not converge. A recent work [5] demonstrates that using randomization in the recommendation function $\mathcal{R}$ in a non-trivial manner can break this divergence. Finding a reasonable way to do so (in terms of social welfare) in our model is left as an open question. Third, implementing cooperation using other solution concepts like no-regret learning and correlated or coarse-correlated equilibrium are also natural extensions of this work. Lastly, our modeling neglects many real-world aspects of RSs: Providers join and leave the system, demand for content changes over time, providers create content of several types, etc. Future work with a more complex modeling is required for implementing our ideas in real-world applications.

## Broader Impact

It is well-understood in the Machine Learning community that economic aspects must be incorporated into machine learning algorithms. In that view, estimating content satisfaction in RSs is not enough. As we argue in this paper, content providers depend on the system for some part of their income; thus, their better treatment makes them the main beneficiaries of the stance this paper offers. We envision that RSs that will coordinate their content providers (and hence the content available for recommendation) will suffer from less fluctuations, be deemed fairer by all their stakeholders, and will enjoy long-term consumer engagement.

## Acknowledgements

We thank the anonymous reviewers for providing helpful and insightful comments. The work of O. Ben-Porat is partially funded by a PhD fellowship from JPMorgan Chase & Co. The work of M. Tennenholtz is funded by the European Research Council (ERC) under the European Union's Horizon 2020 research and innovation programme (grant agreement n° 740435).

## Footnotes

[1]We use the term *utility* to address the well-being of the content providers, and *social welfare* for the well-being of the content consumers.

[2]There are some exceptions, e.g., YouTube instructing providers how to find their niche [1]. However, these are sporadic, primitive, and certainly do not enjoy recent technological advancements like collaborative filtering.

[3]We do not say that the RSs should dictate authors what to write; instead, it should suggest to each author profitable topics that he/she can write on competently to increase her utility.

[4]We use authors and players interchangeably.

[5]In case no author writes on topic $k$ under $\boldsymbol{a}$, $\mathcal{R}$ do not make any recommendation. As reflected in the utility function $\mathcal{U}$ through the indicator $\mathbb{1}_{a_j=k}$, readers associated with a non-selected topic $k$ do not contribute to any author's utility.

[6]An accessible version of Figure 1 appears in the appendix.

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
