[Supplementary Material]

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

\boldsymbol{a}^1 = (2,1,1) & \boldsymbol{a}^2 = (3,1,1) & \boldsymbol{a}^3 = (4,1,1) & \boldsymbol{a}^4 = (5,1,1) & \boldsymbol{a}^5 = (5,5,1) & \boldsymbol{a}^6 = (1,5,1) \\
\boldsymbol{a}^7 = (2,5,1) & \boldsymbol{a}^8 = (3,5,1) & \boldsymbol{a}^9 = (4,5,1) & \boldsymbol{a}^{10} = (4,4,1) & \boldsymbol{a}^{11} = (1,4,1) & \boldsymbol{a}^{12} = (2,4,1) \\
\boldsymbol{a}^{13} = (3,4,1) & \boldsymbol{a}^{14} = (3,3,1) & \boldsymbol{a}^{15} = (1,3,1) & \boldsymbol{a}^{16} = (2,3,1) & \boldsymbol{a}^{17} = (2,2,1) & \boldsymbol{a}^{18} = (1,2,1) \\
\boldsymbol{a}^{19} = (1,2,2) & \boldsymbol{a}^{20} = (1,1,2) & \boldsymbol{a}^{21} = (3,1,2) & \boldsymbol{a}^{22} = (4,1,2) & \boldsymbol{a}^{23} = (5,1,2) & \boldsymbol{a}^{24} = (5,5,2) \\
\boldsymbol{a}^{25} = (1,5,2) & \boldsymbol{a}^{26} = (3,5,2) & \boldsymbol{a}^{27} = (4,5,2) & \boldsymbol{a}^{28} = (4,4,2) & \boldsymbol{a}^{29} = (1,4,2) & \boldsymbol{a}^{30} = (3,4,2) \\
\boldsymbol{a}^{31} = (3,3,2) & \boldsymbol{a}^{32} = (1,3,2) & \boldsymbol{a}^{33} = (1,3,3) & \boldsymbol{a}^{34} = (1,1,3) & \boldsymbol{a}^{35} = (4,1,3) & \boldsymbol{a}^{36} = (5,1,3) \\
\boldsymbol{a}^{37} = (5,5,3) & \boldsymbol{a}^{38} = (1,5,3) & \boldsymbol{a}^{39} = (4,5,3) & \boldsymbol{a}^{40} = (4,4,3) & \boldsymbol{a}^{41} = (1,4,3) & \boldsymbol{a}^{42} = (1,4,4) \\
\boldsymbol{a}^{43} = (1,1,4) & \boldsymbol{a}^{44} = (5,1,4) & \boldsymbol{a}^{45} = (5,5,4) & \boldsymbol{a}^{46} = (1,5,4) & \boldsymbol{a}^{47} = (1,5,5) & \boldsymbol{a}^{48} = (1,1,5)
\end{array}
$$

Figure 1: A long improvement path for the instance in the proof sketch of Theorem 2.

for $c = \frac{1}{PT}$. The first column of the matrix, which is associated with the quality of topic 1, is identical for all authors. The snake-shape path in the matrix is always greater than the value $c$ in the first column, and is monotonically increasing (top-down). The immediate implications are a) odd players improve their quality when deviating to a topic with a greater index, while even players improve their quality when deviating to a topic with a smaller index (which is not topic 1); and b) every player is more competent than all the players that precede her on every topic but topic 1. The initial profile is $\boldsymbol{a}^0 = (1,1,\dots,1)$. We construct the BRD that appears in Figure 1.[6] It comprises three types of steps: Purple, green and yellow. In purple steps, author 1 deviates to a topic with a higher index. In yellow steps, author 2 deviates to the topic selected by author 1 (e.g., in $\boldsymbol{a}^5$) or author 3 deviates to the topic selected by author 2 (e.g., in $\boldsymbol{a}^{19}$). Green steps always follow yellow steps. In green steps, the author whose topic was selected in the previous step by an author with a higher index deviates back to topic 1 (e.g., author 1 in $\boldsymbol{a}^6$ after author 2 selects topic 5 in $\boldsymbol{a}^5$, or author 2 in $\boldsymbol{a}^{20}$ after author 3 selects topic 2 in $\boldsymbol{a}^{19}$).

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

[9]In fact, any monotonically increasing values along the snake (top-down) will suffice; these are selected for readability.

[10] https://en.wikipedia.org/wiki/Binomial_coefficient#Multiset_(rising)_binomial_coefficient.

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

| Symbol | Description | First introduced |
|---|---|---|
| $\mathcal{P}$ | set of players, $\mathcal{P} = \{1, 2, \ldots P\}$ | Section 2 |
| $P$ | number of players, $P = |\mathcal{P}|$ | Section 2 |
| $\mathcal{T}$ | set of players, $\mathcal{T} = \{1, 2, \ldots T\}$ | Section 2 |
| $T$ | number of topics, $T = |\mathcal{T}|$ | Section 2 |
| $\boldsymbol{a}$ | strategy profile, $\boldsymbol{a} = (a_1, a_2, \ldots a_P)$ | Section 2 |
| $j$ | player index, $j \in \mathcal{P}$ | Section 2 |
| $k$ | topic index, $k \in \mathcal{T}$ | Section 2 |
| $\mathcal{Q}$ | quality matrix, $\mathcal{Q}_{j,k}$ is the quality of player $j$ on topic $k$ | Section 2 |
| $\mathcal{D}$ | user mass over topics, $\mathcal{D}(k)$ is the demand for topic $k$ | Section 2 |
| $\mathcal{R}$ | Recommendation function | Section 2 |
| $\mathcal{C}$ | conversion matrix, $\mathcal{C}_{j,k}$ is the conversion of player $j$ on topic $k$ | Section 2 |
| $\mathcal{U}$ | utility function, $\mathcal{U}_j(\boldsymbol{a})$ is player $j$'s utility under profile $\boldsymbol{a}$ | Section 2 |
| $B_k(\boldsymbol{a})$ | highest quality of blog written on $k$ under profile $\boldsymbol{a}$ | Section 2 |
| $H_k(\boldsymbol{a})$ | number of top-quality bloggers on topic $k$ under profile $\boldsymbol{a}$ $\mathcal{R}$ | Section 2 |
| $G = (X \cup Y, E)$ | bipartite graph with parts $X$ and $Y$ | Section 4 |
| $W$ | subset of nodes in a graph (in the context of saturated sets) | Section 4 |
| $A_k$ | set of high-quality authors on topic $k$ | Algorithm 1 |
| $L_k$ | load on topic $k$ | Algorithm 1 |
| $M$ | partial matching in a graph | Algorithm 1 |
| $Z_k(\boldsymbol{a})$ | highest conversion of displayed content on topic $k$ under profile $\boldsymbol{a}$ | Section B |
| $\gamma$ | an improvement path, better-respond dynamic | Section B |
| $\underline{H_k}$ | minimal $H_k$ value throughout the path $\gamma$ | Section B |
| $\overline{B_k}$ | maximal $B_k$ value throughout the path $\gamma$ | Section B |
| $\overline{Z_k}(\gamma)$ | maximal $Z_k$ value throughout the path $\gamma$ | Section B |
| $p$ | parameter of Recurse, indicates the number of players | Section C |
| $S$ | parameter of Recurse, indicates the available topics | Section C |

Figure 2: Notation table.

# A    Omitted Claims from Section 5

## A.1    The Value of Coordination

Many worst-case measures of the inefficiency due to selfish behavior were proposed over the years, e.g., the Price of Anarchy [17, 43]. In this work, however, we care about social welfare and the impact of intervention in the dynamic (that we introduce in Section 3); thus, to distill the lift in social welfare due to coordination, we focus on the Price of Correlation [3].

**Definition 2.** *Given a game instance, the Price of Correlation is* $PoC \overset{\text{def}}{=} \frac{\max_{e \in E} SW(e)}{\min_{e' \in E} SW(e')}$, *where* $E$ *is the set of PNE profiles.*

**Proposition 1.** *The price of Correlation can be unbounded.*

*Proof.* The construction relies on the tension between content quality and conversion. Consider the following two-player ($P = 2$) with two topics ($T = 2$). Let the demand distribution $\mathcal{D}$ such that $\mathcal{D}(1) = 1 - \epsilon, \mathcal{D}(2) = \epsilon$, and let the quality and conversion matrices be

$$\mathcal{Q} = \begin{pmatrix} 1 & 1 \\ \epsilon & 1 \end{pmatrix}, \qquad \mathcal{C} = \begin{pmatrix} \epsilon/1-\epsilon & 1 \\ \epsilon & 1 \end{pmatrix}.$$

The normal-form game resulting from this bi-matrix is

$$\begin{array}{cc} & \begin{array}{cc} \text{topic 1} & \quad \text{topic 2} \end{array} \\ \begin{array}{c} \text{topic 1} \\ \text{topic 2} \end{array} & \left[ \begin{array}{cc} \epsilon, 0 & \epsilon, \epsilon \\ \epsilon, (1-\epsilon)\epsilon & \frac{\epsilon}{2}, \frac{\epsilon}{2} \end{array} \right] \end{array}$$

It is immediate to see that both profiles $(1, 2)$ and $(2, 1)$ are in equilibrium. However, $SW(1, 2) = 1$, while $SW(2, 1) = \epsilon(1 - \epsilon) + \epsilon < 2\epsilon$. The result is obtained when taking $\epsilon$ to zero. □

## A.2 Relaxing Assumption 1

**Proposition 2.** *If Assumption 1 does not hold, there can be infinite BRDs.*

*Proof.* It suffices to show a better-response cycle. Consider the following three-player ($P = 3$) with three topics ($T = 3$). Let the demand distribution $\mathcal{D}$ such that $\mathcal{D}(1) = \mathcal{D}(2) = \mathcal{D}(3) = d = \frac{1}{3}$, and let the quality and conversion matrices be

$$\mathcal{Q} = q \begin{pmatrix} 1 & 10 & 10 \\ 10 & 1 & 0 \\ 0 & 5 & 5 \end{pmatrix}, \qquad \mathcal{C} = c \begin{pmatrix} 10 & 1 & 5 \\ 1 & 10 & 0 \\ 0 & 1 & 5 \end{pmatrix}.$$

for $0 < q \leq \frac{1}{10}$ and $0 < c \leq \frac{1}{30}$. From here on, since the utilities are linear in the term $cdq$, we omit it from the analysis. Let $\boldsymbol{a}^0 = (2, 1, 3)$. Consider the following sequence of better improvements:

1. $\boldsymbol{a}^1 = (3, 1, 3)$. Player 1 is the deviating one, and $1 = \mathcal{U}_1(\boldsymbol{a}^0) < \mathcal{U}_1(\boldsymbol{a}^1) = 5$.

2. $\boldsymbol{a}^2 = (3, 2, 3)$. Player 2 is the deviating one, and $1 = \mathcal{U}_2(\boldsymbol{a}^1) < \mathcal{U}_2(\boldsymbol{a}^2) = 10$.

3. $\boldsymbol{a}^3 = (3, 2, 2)$. Player 3 is the deviating one, and $0 = \mathcal{U}_3(\boldsymbol{a}^2) < \mathcal{U}_3(\boldsymbol{a}^3) = 1$.

4. $\boldsymbol{a}^4 = (1, 2, 2)$. Player 1 is the deviating one, and $5 = \mathcal{U}_1(\boldsymbol{a}^3) < \mathcal{U}_1(\boldsymbol{a}^4) = 10$.

5. $\boldsymbol{a}^5 = (1, 1, 2)$. Player 2 is the deviating one, and $0 = \mathcal{U}_2(\boldsymbol{a}^4) < \mathcal{U}_2(\boldsymbol{a}^5) = 1$.

6. $\boldsymbol{a}^6 = (2, 1, 2)$. Player 1 is the deviating one, and $0 = \mathcal{U}_1(\boldsymbol{a}^5) < \mathcal{U}_1(\boldsymbol{a}^6) = 1$.

7. $\boldsymbol{a}^7 = (2, 1, 3)$. Player 3 is the deviating one, and $0 = \mathcal{U}_3(\boldsymbol{a}^6) < \mathcal{U}_3(\boldsymbol{a}^7) = 5$.

Notice that $\boldsymbol{a}^0 = \boldsymbol{a}^7$, hence this is indeed an improvement cycle. $\qquad \square$

# B  Proof of Theorem 1

In this section, we formally prove Theorem 1. We begin by setting a few convenient notations in Subsection B.1, prove some useful claims in Subsection B.2, and finally prove the theorem in Subsection B.3.

## B.1  Notations for this Section

In addition to $H_k(\boldsymbol{a})$ and $B_k(\boldsymbol{a})$ introduced in Section 2, we also let

$$Z_k(\boldsymbol{a}) = \max_{1 \leq j \leq n} \{\mathbb{1}_{j \in H_k(\boldsymbol{a})} \mathcal{C}_{j,k}\}.$$

In words, $Z_k(\boldsymbol{a})$ is the highest conversion of a player writing on topic $k$ under the strategy profile $\boldsymbol{a}$. We denote an improvement path, i.e., a sequence of improvement profiles by $\gamma$. For a given $\gamma$, we use $\overline{B_k}(\gamma) \stackrel{\text{def}}{=} \max_{\boldsymbol{a} \in \gamma} B_k(\boldsymbol{a})$ to denote the highest value of $B_k$ along the profiles on the path $\gamma$. Similarly, we let $\underline{H_k}(\gamma) \stackrel{\text{def}}{=} \min_{\boldsymbol{a} \in \gamma} |H_k(\boldsymbol{a})|$ be the minimal size of $H_k$ along $\gamma$, and $\overline{Z_k}(\gamma) \stackrel{\text{def}}{=} \max_{\boldsymbol{a} \in \gamma} Z_k(\boldsymbol{a})$ be the highest $Z_k$ along $\gamma$.

Finally, when a strategy profile $\boldsymbol{a}^r$ is part of an improvement path, we use $p_r$ to denote the index of the deviating player. Namely $p_r$ is the only player such that $a_{p_r}^r \neq a_{p_r}^{r+1}$.

## B.2  Useful Claims

The following observation is an immediate corollary of Assumption 1.

**Observation 1.** *If $\mathcal{Q}_{k,j_1} = \mathcal{Q}_{k,j_2}$ for some topic $k \in M$ and two players $j_1, j_2 \in N$, then $\mathcal{C}_{k,j_1} = \mathcal{C}_{k,j_2}$*

The next proposition shows that after a deviation to a topic, the highest quality on that topic can only increase.

**Proposition 3.** *Let $\gamma$ be a finite improvement path, and let $a_{p_r}^{r+1} = k$ for an arbitrary improvement step $r$. It holds that $\mathcal{Q}_{p_r,k} \geq B_k(\boldsymbol{a}^r)$.*

**Proof of Proposition 3.** Since author $p_r$ improves her utility, $\mathcal{U}_{p_r}(\boldsymbol{a}^r) < \mathcal{U}_{p_r}(\boldsymbol{a}^{r+1})$. By definition of $\mathcal{R}^{\text{top}}$, if $\mathcal{Q}_{p_r,k} < B_k(\boldsymbol{a}^r)$ then $\mathcal{U}_{p_r}(\boldsymbol{a}^{r+1}) = 0 \leq \mathcal{U}_{p_r}(\boldsymbol{a}^r)$, which results in a contradiction. $\square$

In Proposition 4 we bound the utility of an improving author in an improvement step.

**Proposition 4.** *Let $\gamma$ be a finite improvement path, and let $a_{p_r}^{r+1} = k$ for an arbitrary improvement step $r$. If $\mathcal{Q}_{p_r,k} \leq B_k(\boldsymbol{a}^r)$, then*

$$\mathcal{U}_{p_r}(\boldsymbol{a}^{r+1}) \leq \frac{\mathcal{D}(k) \cdot \overline{Z_k}(\gamma)}{\underline{H_k}(\gamma) + 1}.$$

**Proof of Proposition 4.** We are given that $\mathcal{Q}_{p_r,k} \leq B_k(\boldsymbol{a}^r)$. Combined with Proposition 3, we know that

$$\mathcal{Q}_{p_r,k} = B_k(\boldsymbol{a}^r). \tag{4}$$

Notice that $a_{p_r}^r \neq k$ and $a_{p_r}^{r+1} = k$; hence, together with Equation (4) we obtain

$$|H_k(\boldsymbol{a}^{r+1})| = |H_k(\boldsymbol{a}^r)| + 1 \overset{\text{Def. of } \underline{H_k}(\gamma)}{\geq} \underline{H_k}(\gamma) + 1. \tag{5}$$

Observe that Equation (5) suggests that

$$\mathcal{U}_{p_r}(\boldsymbol{a}^{r+1}) = \frac{\mathcal{D}(k) \cdot \mathcal{C}_{p_r,k}}{|H_k(\boldsymbol{a}^{r+1})|} = \frac{\mathcal{D}(k) \cdot \mathcal{C}_{p_r,k}}{|H_k(\boldsymbol{a}^r)| + 1} \leq \frac{\mathcal{D}(k) \cdot \mathcal{C}_{p_r,k}}{\underline{H_k}(\gamma) + 1} \leq \frac{\mathcal{D}(k) \cdot \overline{Z_k}(\gamma)}{\underline{H_k}(\gamma) + 1},$$

where the last inequality holds since $\mathcal{C}_{p_r,k} \leq Z_k(\boldsymbol{a}^{r+1}) \leq \overline{Z_k}(\gamma)$. This concludes the proof of this proposition. $\square$

**Proposition 5.** *If $c = (\boldsymbol{a}^1, \ldots, \boldsymbol{a}^l = \boldsymbol{a}^1)$ is an improvement cycle and $k$ is a topic such that*

1. *there exists an improvement step $r_1$ satisfying $|H_k(\boldsymbol{a}^{r_1})| \neq |H_k(\boldsymbol{a}^{r_1+1})|$, and*

2. *for every improvement step $r_2$, $B_k(\boldsymbol{a}^{r_2}) = \overline{B_k}(c)$,*

*then there exist an index $r$ such that $a_{p_r}^r = k$ and*

$$\mathcal{U}_{p_r}(\boldsymbol{a}^r) = \frac{\mathcal{D}(k) \cdot \overline{Z_k}(c)}{\underline{H_k}(c) + 1}.$$

**Proof of Proposition 5.** From Property 1 we know that there exists an improvement step $r_1$ such that $|H_k(\boldsymbol{a}^{r_1})| \neq |H_k(\boldsymbol{a}^{r_1+1})|$. Assume w.l.o.g. that $|H_k(\boldsymbol{a}^{r_1})| > |H_k(\boldsymbol{a}^{r_1+1})|$; hence

$$|H_k(\boldsymbol{a}^{r_1})| > |H_k(\boldsymbol{a}^{r_1+1})| \geq \underline{H_k}(c); \tag{6}$$

hence, $|H_k(\boldsymbol{a}^{r_1})| \geq \underline{H_k}(c) + 1$. By definition of $\underline{H_k}(c)$ we know that there exists an improvement step $r_3$ such that

$$|H_k(\boldsymbol{a}^{r_3})| = \underline{H_k}(c). \tag{7}$$

From Property 2 we get that for every improvement step $r_2$, $B_k(\boldsymbol{a}^{r_2}) = \overline{B_k}(c)$, which implies that

$$||H_k(\boldsymbol{a}^{r_2})| - |H_k(\boldsymbol{a}^{r_2+1})|| \leq 1. \tag{8}$$

Moreover, Property 2 along with Assumption 1 imply that

$$\mathcal{C}_{p_{r_2},k} = \overline{Z_k}(c). \tag{9}$$

Combining Equations (6)-(9) with the fact that $c$ is an improvement cycle leads to the existence of an improvement step $r$ such that $\boldsymbol{a}^r \in \{\boldsymbol{a}^{r_1}, \boldsymbol{a}^{r_1+1}, \ldots, \boldsymbol{a}^{r_3-1}\}$, $|H_k(\boldsymbol{a}^r)| = \underline{H_k}(c) + 1$, and $|H_k(\boldsymbol{a}^{r+1})| = \underline{H_k}(c)$. This suggests that $a_{p_r}^r = k$ and $\mathcal{Q}_{p_r,k} = B_k(\boldsymbol{a}^r) = \overline{B_k}(c)$; therefore,

$$\mathcal{U}_{p_r}(\boldsymbol{a}^r) = \frac{\mathcal{D}(k) \cdot \mathcal{C}_{p_r,k}}{|H_k(\boldsymbol{a}^r)|} = \frac{\mathcal{D}(k) \cdot \overline{Z_k}(c)}{\underline{H_k}(c) + 1}.$$

$\square$

### B.3 Proof of Theorem 1

To ease the presentation, throughout this subsection we re-index the topics according to the following order
$$\mathcal{D}(1) \cdot \overline{Z_1}(c) \geq \mathcal{D}(2) \cdot \overline{Z_2}(c) \geq \ldots \geq \mathcal{D}(T) \cdot \overline{Z_T}(c).$$
The proof of Theorem 1 relies on several supporting lemmas, which are proven first.

**Lemma 2.** *If $c = (\boldsymbol{a}^1, \ldots, \boldsymbol{a}^l = \boldsymbol{a}^1)$ is an improvement cycle, then for every improvement step $r$ and every topic $k$ it holds that $B_k(\boldsymbol{a}^r) = B_k(\boldsymbol{a}^{r+1})$.*

**Proof of Lemma 2.** Assume w.l.o.g. that $c$ is a simple improvement cycle. It suffices to show that $B_k(\boldsymbol{a}^r) \leq B_k(\boldsymbol{a}^{r+1})$ for every $r$ and $k$, since this implies
$$B_k(\boldsymbol{a}^1) \leq B_k(\boldsymbol{a}^2) \leq \ldots \leq B_k(\boldsymbol{a}^{l-1}) \leq B_k(\boldsymbol{a}^l) = B_k(\boldsymbol{a}^1).$$
The left-hand-side and the right-hand-side of the inequality above are identical; thus, they must all hold in equality.

We prove by induction on the topic index $k$ that $B_k(\boldsymbol{a}^r) \leq B_k(\boldsymbol{a}^{r+1})$ holds for every $r$, $1 \leq r \leq l-1$.

**Base case** As we elaborate shortly, the base case is a special case of the Step.

**Step** Suppose the assertion holds for every $k$ where $k < K \leq T$, but does not hold for $K$ (where $K = 1$ is the inductive base, for which we assume nothing). For better readability, we divide the analysis into parts.

Part 1: By definition of $\overline{B_K}(c)$, there exists $r'$, $1 \leq r' \leq l-1$ such that $B_K(\boldsymbol{a}^{r'}) = \overline{B_K}(c)$. Since the assertion does not hold for $K$, there exists $r''$, $1 \leq r'' \leq l-1$, such that $B_K(\boldsymbol{a}^{r''}) > B_K(\boldsymbol{a}^{r''+1})$. Therefore, as $c$ is an improvement cycle, there exists $r_1$ such that $\boldsymbol{a}^{r_1} \in \{a^{r'}, a^{r'+1}, \ldots, a^{r''}\}$ and $\overline{B_K}(c) = B_K(\boldsymbol{a}^{r_1}) > B_K(\boldsymbol{a}^{r_1+1})$.

As a result, it holds for the improving author $p_{r_1}$ in step $r_1$ that $\mathcal{Q}_{p_{r_1}, K} = \overline{B_K}(c) > B_K(\boldsymbol{a}^{r_1}_{-p_{r_1}})$ and $|H_K(\boldsymbol{a}^{r_1})| = 1$. Put differently, the quality of author $p_{r_1}$'s document exceeds all other qualities under $\boldsymbol{a}^{r_1}$ on topic K; thus,
$$\mathcal{U}_{p_{r_1}}(\boldsymbol{a}^{r_1}) = \mathcal{D}(K) \cdot \overline{Z_K}(c). \tag{10}$$

In addition, $p_{r_1}$ is the improving author so $\mathcal{U}_{p_{r_1}}(\boldsymbol{a}^{r_1}) < \mathcal{U}_{p_{r_1}}(\boldsymbol{a}^{r_1+1})$; hence, with Equation (10) we get
$$\mathcal{D}(K) \cdot \overline{Z_K}(c) < \mathcal{U}_{p_{r_1}}(\boldsymbol{a}^{r_1+1}). \tag{11}$$

Let $k_1$ denote the topic that author $p_{r_1}$ is writing on under $\boldsymbol{a}^{r_1+1}$, i.e $k_1 = a^{r_1+1}_{p_{r_1}}$. By definition of $\mathcal{U}$ we obtain
$$\mathcal{U}_{p_{r_1}}(\boldsymbol{a}^{r_1+1}) \leq \mathcal{D}(k_1) \cdot \mathcal{C}_{p_{r_1}, k_1} \leq \mathcal{D}(k_1) \cdot \overline{Z_{k_1}}(c). \tag{12}$$

Inequalities (11) and (12) suggest that $\mathcal{D}(K) \cdot \overline{B_K}(c) < \mathcal{D}(k_1) \cdot \overline{B_{k_1}}(c)$ holds. Recall that we re-indexed the topics according to a decreasing order of $D \cdot V$, and hence $k_1 < K$ (for the base case $K = 1$ and thus we get a contradiction). To summarize this part, we conclude that there must exist a topic $k_1$ such that $k_1 < K$ and $\mathcal{D}(k_1) \cdot \overline{Z_{k_1}}(c) \geq \mathcal{D}(K) \cdot \overline{Z_K}(c)$.

Part 2: Since $k_1 < K$, the induction hypothesis hints that $B_{k_1}(\boldsymbol{a}^{r_1}) = B_{k_1}(\boldsymbol{a}^{r_1+1})$; therefore, $\mathcal{Q}_{p_{r_1}, k_1} \leq B_{k_1}(\boldsymbol{a}^{r_1})$ holds and by Proposition 3 we get that $\mathcal{Q}_{p_{r_1}, k_1} = B_{k_1}(\boldsymbol{a}^{r_1})$. By invoking Proposition 4 for $c, r_1$ and $k_1$ we get
$$\mathcal{U}_{p_{r_1}}(\boldsymbol{a}^{r_1+1}) \leq \frac{\mathcal{D}(k_1) \cdot \overline{Z_{k_1}}(c)}{\underline{H_{k_1}}(c) + 1}.$$

Together with Inequality (11), we conclude that
$$\mathcal{D}(K) \cdot \overline{Z_K}(c) < \frac{\mathcal{D}(k_1) \cdot \overline{Z_{k_1}}(c)}{\underline{H_{k_1}}(c) + 1}. \tag{13}$$

Next, we wish to find an improvement step such that the improving author's utility strictly bounds the right-hand-side of Inequality (13). Since $a_{p_{r_1}}^{r_1+1} = k_1$ and $\mathcal{Q}_{p_{r_1}, k_1} = B_{k_1}(\boldsymbol{a}^{r_1})$ we get that $|H_{k_1}(\boldsymbol{a}^{r_1})| \neq |H_{k_1}(\boldsymbol{a}^{r_1+1})|$. The inductive assumption suggests that for every improvement step $r'$, $B_{k_1}(\boldsymbol{a}^{r'}) = \overline{B_{k_1}}(c)$; therefore, we can invoke Proposition 5. Proposition 5 guarantees the existence of a step $r_2$ such that $a_{p_{r_2}}^{r_2} = k_1$ and

$$\frac{\mathcal{D}(k_1) \cdot \overline{Z_{k_1}}(c)}{\underline{H_{k_1}}(c) + 1} = \mathcal{U}_{p_{r_2}}(\boldsymbol{a}^{r_2}). \tag{14}$$

Since $p_{r_2}$ is the improving author $\mathcal{U}_{p_{r_2}}(\boldsymbol{a}^{r_2}) < \mathcal{U}_{p_{r_2}}(\boldsymbol{a}^{r_2+1})$ holds, which together with Equation (14) implies

$$\frac{\mathcal{D}(k_1) \cdot \overline{Z_{k_1}}(c)}{\underline{H_{k_1}}(c) + 1} < \mathcal{U}_{p_{r_2}}(\boldsymbol{a}^{r_2+1}). \tag{15}$$

Let $a_{p_{r_2}}^{r_2+1} = k_2$. By definition of $\mathcal{U}$, we know that

$$\mathcal{U}_{p_{r_2}}(\boldsymbol{a}^{r_2+1}) \leq \mathcal{D}(k_2) \cdot \overline{Z_{k_2}}(c). \tag{16}$$

The crucial observation is that $k_2 < K$ must hold. To see this, assume otherwise that $k_2 \geq K$, and $\mathcal{D}(k_2) \cdot \overline{Z_{k_2}}(c) \leq \mathcal{D}(K) \cdot \overline{Z_K}(c)$ follows for the re-indexing of topics. Incorporating Inequalities (13), (15) and (16) we obtain

$$\mathcal{D}(K) \cdot \overline{Z_K}(c) < \frac{\mathcal{D}(k_1) \cdot \overline{Z_{k_1}}(c)}{\underline{H_{k_1}}(c) + 1} < \mathcal{U}_{p_{r_2}}(\boldsymbol{a}^{r_2+1}) \leq \mathcal{D}(k_2) \cdot \overline{Z_{k_2}}(c) \leq \mathcal{D}(K) \cdot \overline{Z_K}(c),$$

which is a contradiction; hence, $k_2 < K$.

To complete this step, notice that the iductive hypothesis suggests that $B_{k_2}(\boldsymbol{a}^{r_2}) = B_{k_2}(\boldsymbol{a}^{r_2+1})$, implying $\mathcal{Q}_{p_{r_2}, k_2} \leq B_{k_2}(\boldsymbol{a}^{r_2})$. By invoking Proposition 4 for $c, r_2$, and $k_2$ we conclude that

$$\mathcal{U}_{p_{r_2}}(\boldsymbol{a}^{r_2+1}) \leq \frac{\mathcal{D}(k_2) \cdot \overline{Z_{k_2}}(c)}{\underline{H_{k_2}}(c) + 1}.$$

Together with Inequality (15), we conclude that

$$\frac{\mathcal{D}(k_1) \cdot \overline{Z_{k_1}}(c)}{\underline{H_{k_1}}(c) + 1} < \frac{\mathcal{D}(k_2) \cdot \overline{Z_{k_2}}(c)}{\underline{H_{k_2}}(c) + 1}. \tag{17}$$

To summarize this part, we conclude that there must exist a topic $k_2$, $k_2 < K$ and $k_2 \neq k_1$ that satisfies Inequality (17).

Part 3: We repeat the process in Part 2 to obtain additional topics $k_3, k_4, \ldots, k_K$, such that for all $i \in [K]$, $k_i < K$ and

$$\frac{\mathcal{D}(k_1) \cdot \overline{Z_{k_1}}(c)}{\underline{H_{k_1}}(c) + 1} < \frac{\mathcal{D}(k_2) \cdot \overline{Z_{k_2}}(c)}{\underline{H_{k_2}}(c) + 1} < \frac{\mathcal{D}(k_3) \cdot \overline{Z_{k_3}}(c)}{\underline{H_{k_3}}(c) + 1} < \ldots < \frac{\mathcal{D}(k_K) \cdot \overline{Z_{k_K}}(c)}{\underline{H_{k_K}}(c) + 1}.$$

While the inequality above contains $K$ elements, there are only $K - 1$ topics with index lower than $K$; hence, at least two of them must be identical, and we obtain a contradiction. We conclude that $B_K(\boldsymbol{a}^r) \leq B_K(\boldsymbol{a}^{r+1})$ for every step $r$. This completes the proof of the induction. $\square$

In addition,

**Lemma 3.** *If $c = (\boldsymbol{a}^1, \ldots, \boldsymbol{a}^l = \boldsymbol{a}^1)$ is an improvement cycle, then for every improvement step $r$ and topic $k$ such that $a_{p_r}^{r+1} = k$ there exist $(r', k')$ such that $a_{p_{r'}}^{r'+1} = k'$ and*

$$\frac{\mathcal{D}(k) \cdot \overline{Z_k}(c)}{\underline{H_k}(c) + 1} < \frac{\mathcal{D}(k') \cdot \overline{Z_{k'}}(c)}{\underline{H_{k'}}(c) + 1}.$$

**Proof of Lemma 3.** Let $r, k$ such that $a_{p_r}^{r+1} = k$. From Lemma 2, we know that for every improvement step $r''$, $B_k(\boldsymbol{a}^{r''}) = \overline{B_k}(c)$; thus, $\mathcal{Q}_{p_r,k} \leq B_k(\boldsymbol{a}^r)$ which by Proposition 3 leads to

$$\mathcal{Q}_{p_r,k} = B_k(\boldsymbol{a}^r) = \overline{B_k}(c). \tag{18}$$

By definition of improvement step, $a_{p_r}^r \neq k$; hence, together with Equation (18), we get that $|H_k(\boldsymbol{a}^r)| \neq |H_k(\boldsymbol{a}^{r+1})|$. Notice that $c$ is a finite improvement path, and that the conditions of Proposition 5 holds; thus, by invoking it for $c, r$, and $k$ we conclude the existence of an index $r'$ such that $a_{p_{r'}}^{r'} = k$ and

$$\frac{\mathcal{D}(k) \cdot \overline{Z_k}(c)}{\underline{H_k}(c) + 1} = \mathcal{U}_{p_{r'}}(\boldsymbol{a}^{r'}).$$

In addition, $p_{r'}$ is the improving author, and so

$$\frac{\mathcal{D}(k) \cdot \overline{Z_k}(c)}{\underline{H_k}(c) + 1} = \mathcal{U}_{p_{r'}}(\boldsymbol{a}^{r'}) < \mathcal{U}_{p_{r'}}(\boldsymbol{a}^{r'+1}). \tag{19}$$

Clearly, $a_{p_{r'}}^{r'+1} = k' \neq k$. Lemma 2 indicates that $B_{k'}(\boldsymbol{a}^{r'}) = B_{k'}(\boldsymbol{a}^{r'+1})$; hence, $\mathcal{Q}_{p_{r'},k'} \leq B_{k'}(\boldsymbol{a}^{r'})$. Having showed the condition of Proposition 4 holds, we invoke it for $r', k'$ and conclude that

$$\mathcal{U}_{p_{r'}}(\boldsymbol{a}^{r'+1}) \leq \frac{\mathcal{D}(k') \cdot \overline{Z_{k'}}(c)}{\underline{H_{k'}}(c) + 1}.$$

Combining the above inequality with Inequality (19), we have

$$\frac{\mathcal{D}(k) \cdot \overline{Z_k}(c)}{\underline{H_k}(c) + 1} < \frac{\mathcal{D}(k') \cdot \overline{Z_{k'}}(c)}{\underline{H_{k'}}(c) + 1}.$$

$\square$

We are now ready to prove Theorem 1.

**Proof of Theorem 1.** Let $\gamma$ be any arbitrary improvement path. Since there is a finite number of different strategy profiles, $\gamma$ can only be infinite if it contains cycles. Assume by contradiction that $\gamma$ contains an improvement cycle $c = (\boldsymbol{a}^1, \boldsymbol{a}^2, \ldots, \boldsymbol{a}^l = \boldsymbol{a}^1)$. Let $r_1$ be an arbitrary improvement step and denote by $k_1$ the topic such that $a_{p_{r_1}}^{r_1+1} = k_1$. From Lemma 3 we know that there exist $(r_2, k_2)$ such that $a_{p_{r_2}}^{r_2+1} = k_2$ and

$$\frac{\mathcal{D}(k_1) \cdot \overline{Z_{k_1}}(c)}{\underline{H_{k_1}}(c) + 1} < \frac{\mathcal{D}(k_2) \cdot \overline{Z_{k_2}}(c)}{\underline{H_{k_2}}(c) + 1}.$$

Since $a_{p_{r_2}}^{r_2+1} = k_2$, we can now use Lemma 3 again in order to find $(r_3, k_3)$ such that $a_{p_{r_3}}^{r_3+1} = k_3$ and

$$\frac{\mathcal{D}(k_2) \cdot \overline{Z_{k_2}}(c)}{\underline{H_{k_2}}(c) + 1} < \frac{\mathcal{D}(k_3) \cdot \overline{Z_{k_3}}(c)}{\underline{H_{k_3}}(c) + 1}.$$

This process can be extended to achieve additional $k_4, k_5, \ldots, k_{T+1}$ such that

$$\frac{\mathcal{D}(k_1) \cdot \overline{Z_{k_1}}(c)}{\underline{H_{k_1}}(c) + 1} < \frac{\mathcal{D}(k_2) \cdot \overline{Z_{k_2}}(c)}{\underline{H_{k_2}}(c) + 1} < \ldots < \frac{\mathcal{D}(k_{T+1}) \cdot \overline{Z_{k_{T+1}}}(c)}{\underline{H_{k_{T+1}}}(c) + 1}.$$

Since there are only $T$ topics while the inequality above contains $T + 1$ elements, there are at least two elements which are identical; thus, we obtain a contradiction. We conclude that an improvement cycle can not exist. The above suggests that every better-response dynamics must converge. $\square$

## C Proof of Theorem 2

In this section, we construct the exponential improvement path stated in Theorem 2 and exemplified in its proof sketch. We first construct the game formally for every $\mathcal{P}$ and $\mathcal{T}$. Second, we define the path recursively via Algorithm 2, and exemplify it using a simple game. Third, we demonstrate several properties of the constructed path, among them its exponential length.

Figure 3: Illustration of the quality matrix for $P = 5$ players and $T = 7$ topics.

---

**Algorithm 2:** Constructing Exponential Better-Response Dynamics

```
1  a ← (1, . . . 1) // initial profile, exists globally and is accessed by Recurse
2  p ← P
3  S ← T \ {1} // S is the list of all non-tie topics
4  Recurse(p, S)
   procedure Recurse(p, S):
5      if p == 1 then
          /* The base case concerns with player 1 only.  */
6          while S ≠ ∅ do
7              a₁ ← min(S) // Player 1 advances
8              S.remove(min(S))
9          return
10     while S ≠ ∅ do
11         execute Recurse(p − 1, S) // See Proposition 6 for the intermediate profile
12         if p is even then
13             aₚ ← max(S) // Player p dislodges
14             S.remove(max(S))
15         else if p is odd then
16             aₚ ← min(S) // Player p dislodges
17             S.remove(min(S))
18         aₚ₋₁ ← 1 // Player p − 1 withdraws
19     return
```

## C.1  Game Construction

Let $P$ and $T$ denote $|\mathcal{P}|$ and $|\mathcal{T}|$, respectively. Let $c = \frac{1}{PT}$, and consider the quality matrix $\mathcal{Q}$ such that

$$\mathcal{Q}_{j,k} = \begin{cases} c & \text{if } k = 1 \\ c \cdot \left(2\left(T-1\right)\left(\frac{j-1}{2}\right) + k\right). & \text{if } k > 1 \text{ and odd } j \\ c \cdot \left(2T + 1 - k + 2\left(T-1\right)\left(\frac{j}{2}-1\right)\right) & \text{if } k > 1 \text{ and even } j \end{cases} \quad (20)$$

Despite the involved definition, $\mathcal{Q}$ has a simple structure; see Figure 3 for illustration. Notice that all players have the same quality w.r.t. topic 1. Moreover, the rest of the qualities (of topics 2 to $T$) follow a snake-shape increment. A structural property of this increment is dominance: Every player $j$ for $1 < j \leq T$ is better than player $j - 1$ on all topics but topic 1.[9]

We define the conversion matrix $\mathcal{C}$ to be identical to $\mathcal{Q}$, $\mathcal{C} = \mathcal{Q}$ (the action-target utility of Ben-Porat et al. [9]). To conclude the game description, let $\mathcal{D}$ be the uniform distribution over $\mathcal{T}$, i.e., $\mathcal{D}(k) = \frac{1}{T}$ for every $k \in \mathcal{T}$. In the next subsection, we construct an exponentially long path in the game $\langle \mathcal{P}, \mathcal{T}, \mathcal{D}, \mathcal{Q}, \mathcal{C} \rangle$.

## C.2 Recursive Path

We define the path recursively over the players and the topics via the procedure `Recurse`, which is detailed in Algorithm 2. We first describe the course of the algorithm and the way `Recurse` operations, and then illustrate it using the example from Subsection 3.2.

In Line 1 we set up the initial profile, which is $(1, 1, \ldots, 1)$. Under this profile, every player gets the same share of the user mass on topic 1, namely $\frac{c}{PT}$. In Line 2 we assign to $p$ the number of players that are still unmatched. In Line 3 we initialize the $S$ to include all the topics but topic 1. Recall that topic 1 is singular, as the qualities of all players are a like. In Line 4 we call the `Recurse` procedure, which is the heart of the construction. Implied by its name `Recurse`. It gets the number of players $p$ and a set of topics $S \subseteq \mathcal{T} \setminus \{1\}$, and makes recursive calls. Every recursive call only concerns with players $1, \ldots, p$ and the topics $S \cup \{1\}$. We now briefly describe the course of its execution, and later elaborate on the path it induces.

Lines 5-9 are devoted for the base case, where $p = 1$. In such a case, player 1 iterates through the topics in $S$ in increasing order; she deviates to topic $\min(S)$, then this topic is removed from $S$, and the while loop in Line 6 continues. When there are no more topics in $S$, the call returns (Line 9).

Otherwise, if $p \geq 2$, we enter the while loop in Line 10. Line 11 includes a recursive call to `Recurse`$(p-1, S)$; this is the only place recursive calls are executed. We then continue according to the parity of $p$. If $p$ is even, we enter the if clause in Line 12. Player $p$ deviates to topic $\max(S)$ (Line 13), and then $\max(S)$ is removed from $S$ (Line 14). Alternatively, if $p$ is odd, we enter the else clause in Line 15. In this case, Player $p$ deviates to topic $\min(S)$ (Line 16), and $\min(S)$ is removed from $S$ (Line 17). The final step of the while loop is the deviation of player $p-1$ to topic 1, in Line 18. When $S$ contains no more topics, the call returns (Line 19).

Having explained the dry details of the procedure, we now get into the crux of `Recurse`$(p, S)$ and the BRD that it forms. When $p$ and $S$ are clear from the contexts, it will be useful to discuss the *partial strategy profile*, addressing players $1, \ldots, p$ only. It is almost straightforward to see that

**Observation 2.** *During the course of* `Recurse`$(p, S)$ *with* $p < P$, *no player* $j$ *with* $j > p$ *plays a topic from* $S$.

Due to this observation, whenever a player deviate to topic different than 1 inside a recursive call, we know that we can neglect players with index higher than $p$ while calculating her utility. To use the recursive construction, we wish to characterize how the partial strategy profile looks like after every call. We will later prove that

**Proposition 6.** *The call* `Recurse`$(p, S)$ *terminates in the partial strategy profile* $\boldsymbol{a}_{p,S}^{\text{even}} \stackrel{\text{def}}{=} (1, 1, \ldots, 1, \min(S))$ *if* $p$ *is even, and* $\boldsymbol{a}_{p,S}^{\text{odd}} \stackrel{\text{def}}{=} (1, 1, \ldots, 1, \max(S))$ *if* $p$ *is odd.*

In words, the call `Recurse`$(p, S)$ terminates when all the players $1, 2, \ldots p-1$ are playing topic 1, while player $p$ plays his best topic from $S$: Either topic $\min(S)$ or $\max(S)$, depending on her index parity. To illustrate, consider the call the quality matrix in Figure 3 and the call `Recurse`$(4, \{2, 3, 4, 5\})$. Since $p = 4$ is even, by the end of this call the partial strategy profile for players 1 to 4 is $(1, 1, 1, 2)$,

Next, we focus on the deviations. Throughout its execution, players deviate using the \"gets" operator, $\leftarrow$, (Lines 7, 13, 16 and 18). Those deviations are always w.r.t. the strategy profile globally defined in Line 1. We prove later that those deviations are in fact improvement steps.

**Proposition 7.** *Throughout the course of* `Recurse`$(p, S)$, *every time a player deviates the deviation is beneficial.*

To simplify the explanation of the procedure, we divide all deviation into three types:

1. *advance* (Line 7): This improvement step is part of the base case, for $p = 1$. Player 1, and only player 1, deviates to the minimal topic in $S$.

2. *dislodge* (Line 13 for even $p$ and 16 for odd $p$): We explain the case for even $p$, and the odd case is similar. After executing `Recurse`$(p-1, S)$ in Line 11, the partial strategy profile of

| Step | Executing call | Advance | Dislodge | Withdraw |
|------|----------------|---------|----------|----------|
| 1 | Recurse$(1, \{2,3,4,5\})$ | →(2,1,1)→(3,1,1)→(4,1,1)→(5,1,1) | | |
| 2 | Recurse$(2, \{2,3,4,5\})$ | | →(5,5,1) | →(1,5,1) |
| 3 | Recurse$(1, \{2,3,4\})$ | →(2,5,1)→(3,5,1)→(4,5,1) | | |
| 4 | Recurse$(2, \{2,3,4\})$ | | →(4,4,1) | →(1,4,1) |
| 5 | Recurse$(1, \{2,3\})$ | →(2,4,1)→(3,4,1) | | |
| 6 | Recurse$(2, \{2,3\})$ | | →(3,3,1) | →(1,3,1) |
| 7 | Recurse$(1, \{2\})$ | →(2,3,1) | | |
| 8 | Recurse$(2, \{2\})$ | | →(2,2,1) | →(1,2,1) |
| 9 | Recurse$(3, \{2,3,4,5\})$ | | →(1,2,2) | →(1,1,2) |
| 10 | Recurse$(1, \{3,4,5\})$ | →(3,1,2)→(4,1,2)→(5,1,2) | | |
| 11 | Recurse$(2, \{3,4,5\})$ | | →(5,5,2) | →(1,5,2) |
| 12 | Recurse$(1, \{3,4\})$ | →(3,5,2)→(4,5,2) | | |
| 13 | Recurse$(2, \{3,4\})$ | | →(4,4,2) | →(1,4,2) |
| 14 | Recurse$(1, \{3\})$ | →(3,4,2) | | |
| 15 | Recurse$(2, \{3\})$ | | →(3,3,2) | →(1,3,2) |
| 16 | Recurse$(3, \{3,4,5\})$ | | →(1,3,3) | →(1,1,3) |
| 17 | Recurse$(1, \{4,5\})$ | →(4,1,3)→(5,1,3) | | |
| 18 | Recurse$(2, \{4,5\})$ | | →(5,5,3) | →(1,5,3) |
| 19 | Recurse$(1, \{4\})$ | →(4,5,3) | | |
| 20 | Recurse$(2, \{4\})$ | | →(4,4,3) | →(1,4,3) |
| 21 | Recurse$(3, \{4,5\})$ | | →(1,4,4) | →(1,1,4) |
| 22 | Recurse$(1, \{5\})$ | →(5,1,4) | | |
| 23 | Recurse$(2, \{5\})$ | | →(5,5,4) | →(1,5,4) |
| 24 | Recurse$(3, \{5\})$ | | →(1,5,5) | →(1,1,5) |

Figure 4: A long improvement path for the illustration in Subsection C.2

players $1, 2, \ldots p - 1$ is $\boldsymbol{a}^{\mathrm{odd}}_{p-1, S}$. Indeed, this is true due to Proposition 6 and $p - 1$ being odd. In particular, player $p - 1$ plays $\max(S)$. When we reach Line 13, player $p$ deviates to $\max(S)$ as well. Recall that by the construction of $\mathcal{Q}$, player $p$'s quality dominates the quality of player $p - 1$ on every topic excluding 1. Consequently, since every topic in $\mathcal{S}$ is always greater than 1, the utility of player $p - 1$ zeros. More pictorially, player $p$ *dislodges* player $p - 1$ from being the highest-quality author on topic $\max(S)$.

3. *withdraw* (Line 18): Player $p - 1$ *withdraw* from writing on a favorable topic (the one she was just dislodged from in Line 13), and deviates to topic 1.

To illustrate the terminology and the path, we return to example proposed in Subsection 3.2, and iterate through the path that the procedure forms. In Figure 4 we give the improvements and the call that executes them.

## C.3   Proofs from Subsection C.2

**Proof of Observation 2.** We prove the claim by induction on the depth of the call stack. If $P = 1$, then the claim holds trivially. To see that the claim holds for $P > 1$, focus on the first call, Recurse$(P, \mathcal{T} \setminus \{1\})$. Recall that the starting profile is $(1, 1, \ldots, 1)$. Since $P > 1$, the procedure enters the while loop in Line 10. Then, the call to Recurse$(P - 1, \mathcal{T})$ is executed (Line 11). W.l.o.g. assume $P$ is even (similarly otherwise), and hence we enter the if condition in Line 12. Player $P$ deviates to $\max(S)$ and dislodges player $P - 1$, and then $\max(S)$ is removed from $S$. Later, in Line 18, player $P - 1$ withdraws to topic 1. In the next iteration of the while loop in Line 10, $S$ do

not contain $a_P$. This reasoning can be applied for every iteration of the while loop during the call $\texttt{Recurse}(P, \mathcal{T})$.

Assume the claim holds for player $J + 1$ and $S \subseteq \mathcal{T}$, and focus on $\texttt{Recurse}(j, S)$ for $j > p$. By the inductive step, we know that players $j + 1, \ldots P$ do not play the topics in $S$. To finalize the proof, we use the same arguments as before to show that $\texttt{Recurse}(j, S)$ only makes calls with $a_j \notin S$. $\square$

**Proof of Proposition 6**. The key ingredient of this proof is to watch the snake-trail closely (see the definition of $\mathcal{Q}$ and Figure 3).

We prove the claim by induction on $p$. First, notice that $S$ is non-empty, because recursive calls can only happen in Line 11, which means the while expression, $S \neq \emptyset$ is true.

The base case is when $p$ equals 1. We enter the if statement in Line 5, and each iteration player 1 *advances* to topic $\min(S)$. By the end of the loop, the only topic still in $S$ is $\max(S)$, player 1 *advances* to $\max(S)$, which is then removed. We are hence guaranteed that at the return command in Line 9, the partial strategy profile for player 1 is $(\max(S))$.

Consider an even $p$ and the call $\texttt{Recurse}(p, S)$, and assume the claim holds for $p - 1$, which is odd. To avoid notational confusion, we will denote by $S'$ the set $S$ at the beginning of the call, and let $S$ change through the course of the call. Therefore, we essentially analyze $\texttt{Recurse}(p, S')$.

Due to the inductive assumption, every time the recursive call to $\texttt{Recurse}(p-1, S)$ for the appropriate subset $S$ returns, the partial strategy profile for players $1, 2, \ldots, p-1$ is $(1, 1, \ldots, 1, \max(S))$. Player $p$ then dislodges player $p - 1$ from $\max(S)$ (Line 13), $\max(S)$ is removed from $S$ and player $p - 1$ withdraws to topic 1 (Line 18). Consequently, by the end last iteration of the while loop in Lines 10, player $p$ dislodges player $p - 1$ from $\min(S')$ (Since $|S| = 1$ and all other topics were removed), the last topic is removed from $S$, and player $p - 1$ withdraws to topic 1; hence, the partial strategy profile for players $1, 2, \ldots, p$ at the end of this call is $(1, 1, \ldots, 1, \min(S'))$ (with $a_{p-1} = 1$ and $a_p = \min(S')$).

Next, consider an odd $p$ and the call $\texttt{Recurse}(p, S)$, and assume the claim holds for $p - 1$, which is even. We follow the same notation convenience as before, and analyze $\texttt{Recurse}(p, S')$.

The arguments are almost identical to the even case, but appear here for completeness. Due to the inductive assumption, every time the recursive call to $\texttt{Recurse}(p - 1, S)$ for the appropriate subset $S$ returns, the partial strategy profile for players $1, 2, \ldots, p - 1$ is $(1, 1, \ldots, 1, \min(S))$. Player $p$ then dislodges player $p - 1$ from $\min(S)$ (Line 16), $\min(S)$ is removed from $S$ and player $p - 1$ withdraws to topic 1 (Line 18). Consequently, by the end last iteration of the while loop in Lines 10, player $p$ dislodges player $p - 1$ from $\max(S')$ (Since $|S| = 1$ and all other topics were removed), the last topic is removed from $S$, and player $p - 1$ withdraws to topic 1; hence, the partial strategy profile for players $1, 2, \ldots, p$ at the end of this call is $(1, 1, \ldots, 1, \max(S'))$ (with $a_{p-1} = 1$ and $a_p = \max(S')$).

This completes the proof of the proposition. $\square$

**Proof of Proposition 7**. We prove the claim by addressing each type of deviation separately.

1. *advance:* (Line 7): Consider the call $\texttt{Recurse}(1, S)$ for some $S$. Due to Proposition 6, the partial profile of player 1 at the beginning of the iteration is $(1)$, namely $a_1 = 1$. Due to Observation 2, no other player plays $\min(S)$, and thus this is a beneficial deviation. Since player 1 advances along the snake-trail in the while loop in Line 6, her deviations are beneficial.

2. *dislodge:* W.l.o.g. consider an even $p$ (Line 13, the odd case is almost identical and hence omitted). Due to Proposition 6, the partial profile of players $1, 2, \ldots, p$ at the beginning of the iteration is $(1, 1, \ldots, 1)$. During the course of the execution, player $p$ deviates only using dislodge operations. In the first iteration of the while loop, player $p$ deviates from topic 1 to topic $\max(S)$. Recall that Observation 2 guarantees that no other player with greater index player $\max(S)$; hence, due to the construction of $\mathcal{Q}$ in Equation (20), she improves her utility from at most $c$ to a strictly greater utility. Afterwards, in every iteration of the while loop, she follows the snake-trail and thus her deviations are beneficial.

3. *withdraw:* (Line 18): We address the case of even $p$, and the odd case follows similarly. After every recursive call the $\texttt{Recurse}(p - 1, S)$, the obtain partial strategy profile is $(1, 1, \ldots, 1, \max(S))$ (partial for players $1, 2, \ldots p - 1$). Recall that player $p$ dislodges player $p - 1$ from her topic, since by the construction of $\mathcal{Q}$, player $p$'s quality dominates the quality of player $p - 1$ on every topic excluding 1. Consequently, the utility of player $p - 1$ zeros. When player $p - 1$ withdraws to topic 1, she gets a strictly positive utility as all the players are of equal quality w.r.t. this topic.

Overall, we have showed that all deviations are beneficial. $\qquad\square$

## C.4   Path Length

We now lower bound the length of the BRD $\texttt{Recurse}$ generates. Let $f(P, s)$ denote the number of profiles $\texttt{Recurse}(P, S)$ iterates (for $s = |S|$). According to the base case (Lines 5-9), if $p = 1$ then $f(1, s) = s$. Furthermore, for completeness, we note that if $s = 0$ then $f(p, 0) = 0$.

For $p, s$ such that $p > 1$ and $s \geq 1$, the analysis should incorporate the deviations in the while loop (Line 10). Every iteration includes a recursive call to $\texttt{Recurse}(p - 1, S')$, were $|S'|$ goes from $s$ to 1 inclusive, and *dislodge* and *withdraw* steps. As a result,

$$f(p, s) = 2s + \sum_{k=1}^{s} f(p - 1, k).$$

Put differently,

$$f(p, s) - f(p, s - 1) = 2s + \sum_{k=1}^{s} f(p - 1, k) - 2(s - 1) - \sum_{k=1}^{s-1} f(p - 1, k) = 2 + f(p - 1, s);$$

therefore,

$$f(p, s) = f(p, s - 1) + f(p - 1, s) + 2.$$

One can solve this recurrence using generating functions, but here we show a much simpler solution. Let $F(p, s)$ denote the *multiset coefficient*[10], i.e., $F(p, s) \stackrel{\text{def}}{=} \left(\!\!\binom{s}{p}\!\!\right) = \binom{p+s-1}{p}$. Next, we show that $f(p, s) \geq F(p, s)$. This inequality holds as equality for the base cases $(1, s)$ for $s \geq 0$ and $(p, 0)$ for $p \geq 0$. Assume that the $f(p', s') \geq F(p', s')$ whenever either $p' < p$ or $s' < s$. It holds that

$$
\begin{aligned}
f(p, s) &= f(p, s - 1) + f(p - 1, s) + 2 \\
&\geq F(p, s - 1) + F(p - 1, s) + 2 \\
&= \binom{p + s - 2}{p} + \binom{p + s - 2}{p - 1} + 2. \\
&= \binom{p + s - 1}{p} + 2. \\
&= \left(\!\!\binom{s}{p}\!\!\right) + 2. \\
&= F(p, s) + 2,
\end{aligned}
$$

where we use the inductive step and Pascal's triangle; thus, $f(p, s) \geq F(p, s)$ holds. Recall that the initial call (Line 4) is $\texttt{Recurse}(|\mathcal{P}|, \mathcal{T} \setminus \{1\})$; therefore, the number of steps is in this call is $f(P, T - 1)$. Using the relation between $f$ and $F$,

$$f(P, T - 1) \geq F(P, T - 1) = \left(\!\!\binom{T - 1}{P}\!\!\right) = \binom{P + T - 2}{P} \geq \left(\frac{T - 2}{P} + 1\right)^{P}.$$

This concludes the proof of Theorem 2 and this section.

# D    Proofs from Section 4

## D.1    Proof of Theorem 3

Before we begin, we make the following notational remarks. When referring to the value of any object used in Algorithm 1, we use the super script $^{t:e}$ to denote the value of that object at the end of the $t$'th iteration, and $^{t:b}$ to denote the value of that object at the beginning of the $t$'th iteration. For instance, $A_k^{t:b}$ or $k^{*t:e}$. In addition, we denote by $\alpha^t$ the value of the maximum in Line 8 in iteration $t$, i.e.,

$$\alpha^t = \max_{j \in A_{k^{*t:e}}^{t:e}} \frac{\mathcal{D}(k^{*t:e})\mathcal{C}_{j,k^{*t:e}}}{L_{k^{*t:e}}^{t:b} + 1}.$$

The proof of this theorem relies on Propositions 8-10 below; we defer their proofs to Subsection D.2. To claim that the returned profile $\boldsymbol{a}^*$ is a PNE, we first need to show that it is a valid strategy profile.

**Proposition 8.** *Algorithm 1 returns a valid strategy profile.*

Next, we claim that $(\alpha)_{t=1}^P$ is monotone.

**Proposition 9.** *The sequence $(\alpha)_{t=1}^P$ is monotonically non-increasing.*

The next proposition lower bounds the utility of every player by the appropriate value of $\alpha$.

**Proposition 10.** *Let $j$ be an arbitrary player index and $t(j)$ be the index of the iteration player $j$ was matched and removed (Lines 16 and 17). It holds that*

$$\mathcal{U}_j(\boldsymbol{a}^*) \geq \alpha^{t(j)}.$$

Using the above propositions, the proof of Theorem 3 is almost straightforward. Assume by contradiction that the claim does not hold; namely, there is a player $j$ and topics $k, k'$ such that $a_j = k$ but $\mathcal{U}_j(\boldsymbol{a}_{-j}^*, k') > \mathcal{U}_j(\boldsymbol{a}^*)$. Let $t(j)$ denote the iteration when we matched and removed player $j$. In addition, notice that $L_{k'}^{t(j):e}$ denotes the number of players selecting $k'$ under $\boldsymbol{a}^*$; thus, $L_{k'}^{t(j):e} = |H_{k'}(\boldsymbol{a}^*)|$. Observe that $j \in H_k(\boldsymbol{a}_{-j}^*, k')$, since otherwise $\mathcal{U}_j(\boldsymbol{a}_{-j}^*, k') = 0$. By invoking Propositions 9 and 10, we get that

$$\mathcal{U}_j(\boldsymbol{a}^*) \geq \alpha^{t(j)} \geq \frac{\mathcal{D}(k')\mathcal{C}_{j,k'}}{L_{k'}^{t(j):e} + 1} = \frac{\mathcal{D}(k')\mathcal{C}_{j,k'}}{|H_{k'}(\boldsymbol{a}_{-j}^*, k')|} = \mathcal{U}_j(\boldsymbol{a}_{-j}^*, k');$$

thus, we obtained a contradiction.

## D.2    Proofs from Subsection D.1

**Proof of Proposition 8.** To prove the proposition, we need to show that every player is matched. Recall that we assume for simplicity that $P \leq T$ (see Footnote 8), or otherwise we add columns of zero to $\mathcal{Q}$ and $\mathcal{C}$ until $P = T$.

Every time the algorithm matches players to topics (Line 16), it removes a set of players and a set of topics form the graph (Lines 17 and 19). Clearly, the number of topic copies we remove $|W|$ equals the number of players we remove, $|N_G(W)|$; thus, in every iteration $|Y| \leq |\tilde{T}|$. To complete the argument, notice that as long as there are players in $Y$ there are topics in $\mathcal{T}$, and hence we will continue to pick $k^*$ (Line 8), add new nodes to $x$ (Lines 10 and 11), and match them with players in $Y$. ▫

**Proof of Proposition 9.** The sequence of sets $(\tilde{\mathcal{T}}^{t:b})_{t=1}^P$ is monotonically non-increasing, since at the beginning $\mathcal{T}^{1:b} = \mathcal{T}$ and afterwards elements can only be removed (in Line 19). In addition, $\mathcal{D}(k)$ is fixed for every $k \in \mathcal{T}$, and $(L_k^{t:b})_{t=1}^P$ is monotonically non-decreasing for every $k \in \mathcal{T}$. The only tricky part is the conversion $\mathcal{C}_{j,k}$. To illustrate, matching and removing a player might decrease the highest quality on topic $k$, thereby changing $A_k$. This change could increase potentially increase the conversion of the players in $A_k$. However, due to Assumption 1, the conversion of the highest-quality player on a fixed topic is non-increasing as we remove players. As a result, for every $t, 1 \leq t < P$

$$\alpha^t = \max_{j \in A_k^{t:e}} \frac{\mathcal{D}(k)\mathcal{C}_{j,k}}{L_k^{t:b} + 1} \geq \max_{j \in A_k^{(t+1):e}} \frac{\mathcal{D}(k)\mathcal{C}_{j,k}}{L_k^{t:b} + 1} \geq \max_{j \in A_k^{(t+1):e+1}} \frac{\mathcal{D}(k)\mathcal{C}_{j,k}}{L_k^{(t+1):b} + 1} = \alpha^{t+1}.$$

▫

---

**Algorithm 3:** Find saturated set in a bipartite graph

---

**Input:** A bipartite graph $G = (X \cup Y, E)$ satisfying the marriage condition
**Output:** A maximum saturated set in $G$
**procedure** GetSaturated($G$):

1     find an $X$-saturated matching $M$ in $G$
2     denote by $\tilde{Y} \subseteq Y$ the nodes that $M$ does not match
3     set directions to the edges in $E$, such that edges in $M$ are directed from $X$ to $Y$ while the other edges are directed from $Y$ to $X$
4     run a BFS starting from the nodes of $\tilde{Y}$ in the directed graph, traversing each edge only once
5     return the set of nodes in $X$ that were not discovered during the BFS

---

**Proof of Proposition 10.** Let $j$ be an arbitrary player, let $t(j)$ be the iteration number it was matched and removed, and let $k$ denote its strategy, $a_j = k$. Notice that

$$\mathcal{U}_j(\boldsymbol{a}^*) = \frac{\mathcal{D}(k)\mathcal{C}_{j,k}}{|H_k(\boldsymbol{a}^*)|} = \frac{\mathcal{D}(k)\mathcal{C}_{j,k}}{L_k^{t(j):e}} \geq \frac{\mathcal{D}(k^{*t(j)})\mathcal{C}_{j,k^{*t(j)}}}{L_{k^{*t(j)}}^{t(j):b} + 1} = \alpha^{t(j)},$$

where the inequality sign holds due to Proposition 9, and holds as equality if and only if $k = k^{*t(j)}$. This concludes the proof of Proposition 10. □

### D.3   Proof of Lemma 1

We prove the lemma by constructing Algorithm 3 and prove its guarantees. Before we begin, we make the following useful argument.

**Proposition 11.** *Let $G = (X \cup Y, E)$ be a bipartite graph satisfying the marriage condition. If $W_1, W_2, \ldots W_k$ are saturated sets, then $\bigcup_{i=1}^{k} W_i$ is also a saturated set.*

We defer the proof to Subsection D.4. Proposition 11 suggests that it suffices to find all nodes that participate in *any* saturated set, since their union forms the maximum saturated set. One useful notion in the algorithm and its analysis is that of a *crossing path*.

**Definition 3** ($M$-crossing path)**.** *Let $M$ be an $X$-saturating matching in $G$. A path $y, x_1, y_1, \ldots, x_k, y_k, x$ between $y \in Y$ and $x \in X$ is called an $M$-crossing path if for every $i, 1 \leq i \leq k$ it holds that $(x_i, y_i) \in M$.*

Moreover, we leverage the definition of crossing paths to show that

**Proposition 12.** *Let $M$ be any arbitrary $X$-saturating matching in $G$, and let $\tilde{Y}$ denote the nodes in $Y$ that where not matched. There exists an $M$-crossing path from a node $y \in \tilde{Y}$ to a node $x \in X$ if and only if $x$ does not participate in any saturated set in $G$.*

We defer the proof to Subsection D.4. We move to describe the details of the procedure GetSaturated, which is given in Algorithm 3. First, in Line 1, we find an $X$-saturated matching. Since we are given that $G$ satisfies the marriage condition, such a matching is guaranteed to exist. To obtain this matching, we can use, e.g., the Hopcroft–Karp algorithm [25]. Then, in Line 2, we denote by $\tilde{Y}$ the set of nodes that were not matched, all belong to $Y$. We then construct a directed version of the graph, such that all paths from $\tilde{Y}$ are $M$-crossing paths (see Definition 3). The final step is to run a BFS to conclude the reachable nodes from $\mathcal{Y}$.

The correctness of GetSaturated follows immediately from Proposition 12. As for running time considerations, the heaviest operation in terms of run-time is using the Hopcroft–Karp algorithm in Line 1, which takes $O\left(\sqrt{|V|}|E|\right)$ time. Lines 2-5 can be executed in $O(|V| + |E|)$ time.

### D.4   Auxiliary Claims for this Section

**Proof of Proposition 11.** We prove the claim for the case of $k = 2$, and the general case follows by induction.

First, by applying the marriage condition on $W_1 \cup W_2$, we get

$$|W_1 \cup W_2| \leq |N_G(W_1 \cup W_2)|. \tag{21}$$

On the other hand, by applying the marriage condition on $W_1 \cap W_2$ and using the fact that $W_1$ and $W_2$ are saturated, we get

$$|W_1 \cup W_2| = |W_1| + |W_2| - |W_1 \cap W_2| \geq |N_G(W_1)| + |N_G(W_2)| - |N_G(W_1 \cap W_2)|. \tag{22}$$

Due to Proposition 13, $|N_G(W_1 \cap W_2)| \leq |N_G(W_1) \cap N_G(W_2)|$; thus, Inequality (22) implies that

$$|W_1 \cup W_2| \geq |N_G(W_1)| + |N_G(W_2)| - |N_G(W_1) \cap N_G(W_2)| = |N_G(W_1 \cup W_2)|.$$

We get the required result by combining Inequalities (21) and (22). $\qquad\square$

**Proof of Proposition 12.** Denote $M$ and $\tilde{Y}$ as in the statement of the proposition.

Direction $\Rightarrow$: Assume there exists an $M$-crossing path $y, x_1, y_1, \ldots, x_k, y_k, x$ for some $y \in \tilde{Y}$, and w.l.o.g. let it be the shortest path. We need to show that for every $W \subseteq X$ such that $x \in W$, $|W| < |N_G(W)|$.

Let $E'$ denote the edges of that $M$-crossing path, when we add the node matched to $x$, $M(x)$ as the final node. Namely, $E$ contains the edges of the path

$$\overbrace{y, \underbrace{x_1, \overbrace{y_1}, x_2}, y_3}^{\notin M} \ldots \overbrace{x_k, \underbrace{y_k, \overbrace{x}, M(x)}}^{\notin M}$$

Notice that the definition of $M$-crossing path does not require anything from edges $(y_i, x_{i+1})_{i=1}^{k-1}$. Nevertheless, if $(y_i, x_{i+1}) \in M$ then $x_i = x_{i+1}$ must hold, since $(x_i, y_i) \in M$ and $M$ matches $y_i$ only once; but this contradict our assumption that the path is the *shortest* $M$-crossing path.

We claim that $M' = (M \Delta E) \cup E$ is an $X$-saturated matching. Clearly, the degree of every node that does not participate in the edges of $E$ was unchanged. Moreover, excluding $y$ and $M(x)$, every node participates in $E$ twice: Once via an edge that belongs to $M$, and once via an edge that does not; hence, the degree of such nodes in $M'$ is also 1. The degree of $y$ is 1 as it is now matched, and the degree of $M(x)$ is now zero since $M'$ does not match it to any node.

Notice that $x$ has at least two neighbors in $G$, $y$ and $M(x)$. Let $W$ be an arbitrary subset of $X$ such that $x \in W$. It holds that

$$|W| = |M'(W)| < |M'(W)| + |\{M(x)\}| \leq |N_G(W)|.$$

Rearranging, we see that $|W| < |N_G(W)|$; thus, $W$ is not saturated. Since we selected $W$ arbitrarily, we proved that $x$ does not participate in any saturated set.

Direction $\Leftarrow$: Assume that there is no $M$-crossing path from any node in $\tilde{Y}$ to $x$. We need to show that $x$ participates in a saturated set.

Let $W \subset X$ denote the set of nodes reachable via $M$-crossing paths from $\tilde{Y}$, and let $\overline{W}$ be its complementary to $X$, i.e., $\overline{W} = X \setminus W$. In particular, our assumption implies that $x \in \overline{W}$. Observe that $\tilde{Y}, M(W)$ and $M(\overline{W})$ is a partition of $Y$.

We aim to show that $N_G(\overline{W}) = M(\overline{W})$. Indeed, that suffices as by definition of matching, $|M(\overline{W})| = |\overline{W}|$. We claim that every node $y \in N_G(\overline{W})$ must be matched to a node in $\overline{W}$. To see way this is true, assume the converse. Let $x_y$ denote a neighbor of $y$ in $\overline{W}$. If $y \in \tilde{Y}$, then $x_y \in W$ by the way we defined $W$. Otherwise, $y \in M(W)$ (or equivalently, $M(y) \in W$). By definition of $W$, there exists a node $\tilde{y} \in \tilde{Y}$ for which the path

$$\tilde{y}, x_1, y_1, \ldots, x_k, y_k, M(y)$$

is an $M$-crossing path. Since $(M(y), y) \in M$,

$$\tilde{y}, x_1, y_1, \ldots, x_k, y_k, M(y), y, x_y$$

is an $M$-crossing path too; however, this is impossible since it would imply that $x_y \in W$. $\qquad\square$

**Proposition 13.** *Let $G = (X \cup Y, E)$ be a bipartite graph. For every $W_1, W_2 \subseteq X$ it holds that*

$$N_G(W_1 \cap W_2) \subseteq N_G(W_1) \cap N_G(W_2).$$

**Proof of Proposition 13.** For every $v \in N_G(W_1 \cap W_2)$ there exists $u \in W_1 \cap W_2$ such that $v \in N_G(\{u\})$. Since $u \in W_1 \cap W_2$, it follows that $v \in N_G(W_1)$ and $v \in N_G(W_2)$; hence, $v \in N_G(W_1) \cap N_G(W_2)$. □

### D.5  Proof of Corollary 1

Recall the Lemma 1 finds a maximum saturated set in $O(\sqrt{|V|}|E|)$, provided that the graph satisfies the marriage condition.

**Proposition 14.** *Throughout the course of Algorithm 1, every time Line 13 is executed $G$ satisfies the marriage condition.*

**Proof of Proposition 14.** We prove the claim by induction on the iteration number. The marriage condition holds for $t = 1$. At the beginning of the for loop (Line 6), the graph $G$ is empty; after $x^1$ is added to $X$ (Line 11) the only non-empty subset is $\{x^1\}$. Notice that according to the way we pick $k^{*1}$, $x^1$ has at least one neighbor (a player in $A_{k^{*1}}^1$, see Line 7); hence, $\{x_1\} \leq |N_{G^1}(\{x^1\})| = 1$.

Assume the claim holds for iterations $1, \ldots, t - 1$. We distinguish two cases:

1. If a maximum saturated set $W \neq \emptyset$ was discovered in Line 13 of iteration $t - 1$. Due to the inductive step, when executing Line 13 $G$ satisfies the marriage condition. Afterwards, in Lines 15-19 we remove $W$ and $N_G(W)$ altogether. As we show in Proposition 11, every union of saturated sets is also saturated; therefore, by the end of the $t - 1$ for every $W' \subseteq X$, $|W'| < N_G(W')$.

   From that moment on to the time we reach Line 13 in iteration $t$, $X$ is added precisely one node, $x^t$. As a result $|W \cup \{x^t\}| \leq |W| + 1 < |N_G(W)| + 1 \leq |N_G(W \cup \{x^t\})| + 1$, suggesting that $|W \cup \{x^t\}| \leq |N_G(W)|$.

2. Otherwise, the algorithm stepped over the if block of Line 14 in the $t - 1$'th iteration, and continued to the $t$'th iteration. In particular, at the beginning of the $t$'th iteration, we have $|W| < N_G(W)$. From here on we use exactly the same arguments as in the second part of the previous step.

This completes the proof of the proposition. □

Due to Proposition 14, we can use the run-time guarantees of Lemma 1 for finding the maximum saturated set. The total run-time of Lines 7 and 8 is $O(P^{1.5}T)$, as we can sort every column in $\mathcal{Q}$ and remove/add rows as we go. The other Lines, 1-5, 9-12, and 14-20 take at most $O(P + T)$ in every iteration. Finally, notice that for $G$ it holds that $\sqrt{|V|}|E| \leq \sqrt{P \cdot T} \cdot P$; hence, by multiplying it $P$ times (for every iteration of the for loop in Line 6) we get the desired result.