[Reviews · NeurIPS 2020]

Review 1

Summary and Contributions: The authors consider the model of content generation game in a recommender system. The particular model was introduced by Ben Basat et al., JAIR 2017 and has previously been studied in a series of papers by Ben Porat and authors published in NeurIPS, EC and AAAI. The model studied is as follows: It's a game between authors. Each author chooses a topic to write about. They are shown to a viewer randomly if they are one of the highest quality and derive a value equal a conversion factor that depends on the author and the topic. The authors extend previous work by showing that in this game better response dynamics converge under more general conversion structure that satisfies simple monotonicity properties (content of higher quality yields has higher conversion rate). They show that better response dynamics can still take exponential time to converge and provide an algorithm for directly computing a pure Nash equilibrium in these games.

Strengths: Soundness of the claims: The authors justify all their claims with complete proofs. The proofs are involved but very thorough. I was able to follow all the details that I tried to understand. Significance and novelty: The proofs provided by the authors are non-trivial. They require novel ideas and constructions. Relevance to the NeurIPS community: The model studied by the authors has previously been published in JMLR, AAAI, NeurIPS, EC. It also seems to be of practical relevance as it captures the natural competition between content creators on platforms like Medium.

Weaknesses: Significance and novelty: Considering this work by itself, I wonder why pure Nash equilibria, better response dynamics are the main concepts to study in this model. Would no-regret learning and correlated/coarse-correlated equilibrium be a more natural model. Isn't this better thought of as a repeated game between content creators? At the least, wouldn't authors choose a mixed strategy where they write about a variety of topics and see how they fair compared to their competition? Perhaps this is covered by previous work but might be useful to repeat, to justify further effort on pure Nash equilibria and better response dynamics. I was also curious about the focus on better response dynamics instead of best response dynamics. In particular since the authors' last result is about computing a pure Nash equilibrium, can the pure Nash equilibrium be reached using best response dynamics? My apologies if there is a theorem that says that consider best response and better response in equivalent. Post-rebuttal: I am still not fully convinced why better response dynamics are the right dynamics to consider. I am curious to know if any best response dynamics is guaranteed to converge in a polynomial number of steps. As for mixed vs pure strategies - the authors are free to choose any topic that gives them best utility so it is still conceivable that they could randomized between a few different topics.

Correctness: Based on the proofs I read in detail they seem correct. I didn't read all the proofs completely.

Clarity: The paper is well written. Between the main paper and the appendix, the authors have include all the details. I think it's great that the authors provided the example for the better response dynamics not converging it gives a better flavor of the construction instead of providing the full details. (which are included in the appendix). They also include a handy table with all the notation to make it easier to follow the proof.

Relation to Prior Work: Authors discuss relationship to prior work when relevant. Perhaps authors should include more details about the results from the closely related work [4, 5, 6, 7] - that could provide further motivation for why the results considered here fit into the broader research agenda.

Reproducibility: Yes

Additional Feedback: * The authors index the topics according to D_i in one place and according to a different metric in another. I hope that doesn't break anything. Could the earlier one be dropped completely? * In Theorem 2, note that if T is a constant then (T-2/P + 1)^P is not exponential in P. I think T has to be poly(P). * In Algo1 change step 10 to create a *new* node x associated with topic k^* * Minor typos:


Review 2

Summary and Contributions: The paper studies the following game that captures the decisions content providers must make in choosing the topics they cover: Each player (content provider) selects from a finite set of topics. If player j writes on topic k, the quality of this content is an exogenously determined parameter q_{jk}. Readers interested in each topic choose the highest quality content provided (with ties broken uniformly at random), and the content providers are rewarded based on the fraction of readers they get, multiplied by a conversion rate that depends on the content provider and the topic. The main results of the paper are the following theoretical results: this game always has a pure-strategy Nash equilibrium, the better-response dynamics always converges but might take exponential time, and there is an efficient algorithm to compute a Nash equilibrium. The game is essentially something in between a congestion game with player-specific payoff functions and a stable marriage game. Given all the previous work on similar settings (in particular the AAAI'19 paper), the results are not surprising, although actually working out all the technical details is non-trivial. It's worth noting that (if I'm not mistaken) when there is no ties in quality parameters, the game is essentially a stable marriage game, and a Nash equilibrium can be found by an author-proposing algorithm. Therefore, all the complications of Algorithm 1 has to do with ties among quality scores. In terms of motivation, the paper falls under the category of theory papers loosely motivated by a practical scenario. In particular, the model is really about how authors choose a topic to write about, and there's no real connection to the recommender systems, since the model assumes a trivial model of recommendation (the recommender system that always picks an item with the highest quality, for known quality scores).

Strengths: Nice theoretical results. The algorithm for computing a NE is non-trivial and interesting.

Weaknesses: The marginal contribution over the previous work (e.g., AAAI'19 paper) is not that substantial. Problem is not very well motivated.

Correctness: I haven't checked the proofs (that are in the appendix), but the results seem reasonable.

Clarity: Yes.

Relation to Prior Work: Acceptable, but I expected more discussion of the connection to congestion games and the deferred acceptance algorithm for stable marriage.

Reproducibility: Yes

Additional Feedback:


Review 3

Summary and Contributions: This paper formalizes a game theoretical scenario between content provides (players) of the recommendation system. With the solution concept of pure Nash equilibrium, the decentralized algorithm, better-response dynamics, is shown to converge in exponential time, while a centralized algorithm is designed to find a PNE efficiently. The convergence result indicates the existence of PNE is such a game, which is important and the centralized algorithm is mainly leveraging the perfect matching to find the corresponding PNE.

Strengths: The game formalization in this paper is quite novel and characterize most properties in practical recommendation system. Most proofs, as far as I checked, are correct and well-written.The topic is relevance to the NeurIPS community, as the authors are trying to justify the significance of the recommendation system.

Weaknesses: The choice of better-response dynamics (BRD) is questionable. BRD is not popular algorithm to find NE, neither efficient nor guaranteed to converge in most games. I believe it is mainly used to prove the existence of PNE through non-existence of improvement cycle in BRD, which is well-written although relatively not novel. Meanwhile, BRD should not also be considered as a practical algorithm that may be adopted by each content provider alone, since BRD still requires much information about the conversion matrix and demand function. In other word, the exponential running time of BRD is expected, and somewhat meaningless.

Correctness: I checked most proofs and they are correct.

Clarity: This paper is well written. The presentation is good and the notations, although complicated, are well explained. All the proofs in the Appendix are easy to follow. The main drawback is related to the social welfare part. It seems that the authors want to emphasize the social welfare varies a lot among multiple PNE, and the coordination (centralized algorithm) helps players to obtain the PNE with better social welfare. On one hand, it is not a good way to define social welfare only w.r.t. to consumers but not including content providers' utility. On the other hand, the concept of "Price of Correlation" is a very good way to explain this motivation, but it is only mentioned in Appendix. And there is indeed no discussion on comparing the social welfare under the PNE obtained by decentralized and centralized algorithms.

Relation to Prior Work: It is clearly discussed how this work differs from previous contributions.

Reproducibility: Yes

Additional Feedback: I think it may be better to provide an intuitive proof for Theorem 3, explaining why the output is an equilibrium, instead of the one for Lemma 1. Some minor issues: Line 146, two players, two topics; Line 256, undefined notation N_G for neighbors. ========================== Regarding author's response: Although my two major concerns, the choice of BRD and the analysis of social welfare, are responsed, I think there is no new point there. Basically, these two issues maybe challenging to address, but indeed reduce this paper's significance. Thus, I shall not change my score.


Review 4

Summary and Contributions: The work studies the game-theoretical dynamics of "ecosystems" such as blogging platforms. On the one hand we have the content producers, that is, bloggers or filmmakers; on the other hand, the content consumers, that is, the final users, who demand different amounts of content on different topics. On top of this there is a recommender system/platform that suggests contents on the basis of their topics and quality. The only actual players are the producers, who try to maximize their profit (for example, the number of readers of their blog). To this end, each producer can adapt the topic of its content, for example by opening a blog on a more profitable subject. But social welfare, measured as the average quality of the blogs, is at stake too. This complex scenario is formalized as a non-cooperative game. The paper gives two main results. First, under natural assumptions, if each producer sequentially makes a move to increase profit, then the game converges to a pure Nash equilibrium, and that this may require a large number of steps. Second, a pure Nash equilibrium for the system can be computed reasonably fast, in polynomial time; this is done in an interesting way by (loosely speaking) computing max-weight perfect matchings.

Strengths: The subject is of interest of the NeurIPS community (although perhaps not exactly central). The work gives a very clear message with two/three main results. The results are nontrivial (the model is complex and not easy to analyse). The fact that the system converges to a pure Nash equilibrium is interesting (it is not obvious that a pure equilibrium strategy exists, unlike for mixed strategies i.e. distributions). The lower bound construction for the convergence time is neat and insightful. The computation of the equilibrium in polynomial time is interesting as well. The work is well presented and pleasant to read. At a higher level, the work sheds light on the interplay between profit maximization (the bloggers' point of view), social welfare maximization (the readers' point of view), and system design (the recommender system's point of view). This is different from the traditional recommender system problem, that is, suggesting relevant content to users, and different techniques are used. One drawback is that the model is complex, but this is not a fault of the paper. Rather, it is necessary in order to model both the content provider side (bloggers) and the users side (readers) while giving a role to the recommender system/platform. A second drawback is that the social welfare "disappears" in the paper, unless I am missing something. That is, the two results of the paper are oblivious to the social welfare of the system. They are a function of the content producers' utilities but not of the average quality of the content. The only relationship is in the (natural) assumption that higher quality carries higher profit everything else being equal. I find this a bit weird given that the paper brings as motivation the study of long-term social welfare in these dynamic systems.

Weaknesses: One drawback is that the model is complex, but this is not a fault of the paper. Rather, it is necessary in order to model both the content provider side (bloggers) and the users side (readers) while giving a role to the recommender system/platform. A second drawback is that the social welfare "disappears" in the paper, unless I am missing something. That is, the two results of the paper are oblivious to the social welfare of the system. They are a function of the content producers' utilities but not of the average quality of the content. The only relationship is in the (natural) assumption that higher quality carries higher profit everything else being equal. I find this a bit weird given that the paper brings as motivation the study of long-term social welfare in these dynamic systems.

Correctness: I could not check the proofs. The proof sketch of the lower bound sounds correct, though.

Clarity: The paper is clear and well polished.

Relation to Prior Work: Yes.

Reproducibility: Yes

Additional Feedback: I have only two specific comments. - L 146: "three" should be "two" - Theorem 2 and places where it is mentioned: the bound is not exponential in P unless T is a function of P. If T is fixed than the bound is O(exp(T)) which is constant in P.

[Author Response · NeurIPS 2020]

We thank the reviewers for their helpful comments. We will address all suggested minor revisions, clarifications,
corrections and typos. Due to space limitations, we comment here on major questions and suggestions.

**Reviewer 1:** We thank you for appreciating the balance between rigor and intuition.

• "why pure Nash equilibria, better response dynamics..." We employ the one-shot model as an approximation of
the (much more) complex prevailing recommendation systems (RSs). Better response dynamics is the common and
arguably the least demanding way to deal with any asynchronous dynamics in multiagent systems. Indeed, this is the
standard practice in economic studies as well as in engineering contexts (e.g., communication and routing in networks).
Of course, dealing with particular learning rules is of great interest but our emphasis when looking at BRD is on the
set of *all* plausible behaviors. Furthermore, bloggers and vloggers focus on one or two niches and do not mix, which
additionally supports our focus on pure fixed points versus mixed strategies (which are common in CE). Given this
comment, we will address it better in the paper. Nevertheless, minimizing regret is a great tool to obtain CE and CCE,
which also capture some form of centrality; we will propose it as future work, thanks.
• "can the pure Nash equilibrium be reached using best response dynamics?" Recall that a best response dynamic
consists of a starting profile and a series of best-response improvements. Our algorithm can be seen as a best-response
dynamic in which all players start from a *null profile*, assigning all players to a factitious topic with zero user mass, and
then determines the order of best-responding. We will add that to the paper, thanks.

**Reviewer 2:** We thank you for appreciating our theoretical results.

• "when there are no ties in quality parameters, the game is essentially a stable marriage game." This is true (and is
mentioned in prior work). We will write this fact explicitly in our paper, thanks.
• "The problem is not very well motivated." Our conceptual takeaway concerns any RS with strategic content
providers. We claim that the system is not just about the short-sighted recommendation, but rather has a mediator role;
hence, the system can (and ideally should) solve some of the market's inefficiencies. We agree that our theoretical
results cannot be applied as-is (as we have many limiting assumptions). Still, we hope the ideas presented in this paper
and our theoretical grounding will be useful to practitioners.
• "The marginal contribution over the previous work (e.g., AAAI'19 paper) is not that substantial." We respectfully
disagree. We deal with the rate of convergence (Theorem 2) and efficient equilibrium computation (Theorem 3), none
of which are even hinted in that AAAI'19 paper. Theorem 1, which we consider a minor contribution of this paper, is
the only result that builds on techniques from prior work (yet significantly extends it). We stress that our techniques for
proving these two theorems are genuine, and did not appear in prior work.
• "more discussion of the connection to congestion games and the deferred acceptance algorithm for stable marriage."
If accepted, the extra page will allow for more elaboration on these works.

**Reviewer 3:** Thanks for seeing our attempt to justify the significance of the mediator role of RSs in a positive light.

• "BRD is not popular algorithm to find NE..." We agree with the reviewer. We analyze BRD as a means of
decentralized computation, where the RS does nothing but matching demand with supply. In the presence of a
centralized planner, the system can be much more efficient. This is one of the main claims of our paper.
• "social welfare only w.r.t. to consumers but not including content providers' utility" We entirely agree. Note that
measuring the overall welfare of RSs with multiple stakeholders is a philosophical and technical challenge that goes
way beyond this paper (for instance, see *Multisided Fairness for Recommendation* by Robin Burke, 2017). Our attempt
is indeed to use the "Price of Correlation," which, if accepted, we intend to move to the main body based on this
suggestion. We will also highlight that we see "comparing the social welfare under decentralized and centralized
algorithms" as a promising direction for future work, thanks.

**Reviewer 4:** We thank you for appreciating our attempt to incorporate profit maximization into RS design.

• "social welfare "disappears" in the paper." As this reviewer mentions, the holy grail of multiple stakeholder RSs
is to balance the utilities of producers and the welfare of consumers. However, coming up with the right mix is a
longstanding challenge in RSs (see *Multisided Fairness for Recommendation* by Robin Burke, 2017) and is way beyond
the scope of our paper. However, our results demonstrate that by steering the market to the 'right' equilibrium, future
RSs could increase both utilities. As this reviewer mentions, our Assumption 1 sheds light on the connection between
utility (producers) and welfare (consumers); we will better emphasize it in the paper, thanks.
• "the paper brings as motivation the study of long-term social welfare in these dynamic systems." We strongly agree.
The main conceptual takeaway is that welfare-driven recommendations (instead of the current, short-sighted approach)
can steer the market to better equilibrium points. However, this is just the tip of the iceberg; as this reviewer and the
other reviewers mention, long-term welfare requires reaching the best equilibrium (the main technical open question of
this paper), as well as other forms of system design. Due to this comment, we will extend the discussion on that, thanks.

[Meta-Review · NeurIPS 2020]

The reviewers agree and myself agree that this is an interesting game theoretic model to study that could appeal to the neurips community and which has been well-studied already. They also agree that the paper provides a solid theoretical contribution. However, there is a strong worry about the motivation behind the choice of best-response dynamics and the restriction to pure equilibria that although tackled in the rebuttal, still not satisfactorily. Given the strong and non-trivial theoretical results and the relevance of the model, I am would recommend acceptance as a poster, but the authors are strongly encouraged to augment their arguments in the paper regarding the justification of their restrictions above, and how they relate to practical relevance.